engineering geology/energy/structural engineering

conglomerate, gravel content, mechanical properties, microfracture evolution, hydraulic fracturing

**Author for correspondence:**
Hongkui Ge
e-mail: gsggwjb@163.com

# Numerical simulation study on the crack propagation of conglomerate

Senlin Luo[1], Hongkui Ge[1,2], Jianbo Wang[1], Wei Zhou[3,4], Yinghao Shen[2], Pengyu Liu[2] and Jiantong Liu[2]

[1]Department of Petroleum Engineering, China University of Petroleum (Beijing) at Karamay, Xinjiang, Karamay 834000, People's Republic of China
[2]Unconventional Oil and Gas Technology Research Institute, China University of Petroleum, Beijing 102249, People's Republic of China
[3]Research Institute of Experiment and Detection of Xinjiang Oilfield Company, Xinjiang, Karamay 834000, People's Republic of China
[4]Xinjiang Laboratory of Petroleum Reserve in Conglomerate, Xinjiang, Karamay 834000, People's Republic of China

JW, 0000-0003-4435-0291

The conglomerate reservoir is rich in oil and gas reserves; however, the gravel's mechanical properties and laws are difficult to gain through laboratory experiments, which furthermore constrain the hydraulic fracturing design. To analyse the failure law of conglomerate, we simulated the uniaxial compression test based on discrete element software PFC2D and analysed the effect of different cementation strength, gravel content and gravel geometry on the rock deformation and failure characteristics. Results show that (i) as the cementation strength decreases, the compressive strength and elasticity modulus both reduce clearly, while the crack shapes get more complex and the critical value is 0.3; (ii) as the gravel content increases, the conglomerate strength first decreases then increases under the influences of cracks bypassing gravels; cementation strength and gravel content of the conglomerate both contribute to the increase in local additional stress, which leads to a series of changes in crack shapes and mechanical properties of the conglomerate. Based on the above research, the conglomerate strength and crack shapes after failure are relatively complex due to the common influence of cementation strength and gravel content. The gravel edge crack caused by stress concentration is the micro-mechanism that affects the conglomerate mechanical properties.

## 1. Introduction

Reservoirs are generally distributed in sandstone, shale, tight sandstone, carbonatite and igneous rocks, whereas few gas and

oil reservoirs are discovered in conglomerate. The conglomerate reservoirs known so far are Spirit River Formation in Wapiti, Canada, Pennsylvanian basal conglomerate in Garfield field, America, conglomerate reservoir in Edvard Grieg field and Daxing conglomerate reservoir in the Langgu Depression of Bohai Bay Basin, China [1–4]. Unfortunately, reserves of these sand conglomerate are not abundant enough to gain a worldwide attention. In recent years, the exploration results from Mahu Sag in Junggar Basin, northwest China, show that, high-quality oil is found rich in the uppermost permian Urho Formation and lowermost triassic Baikouquan Formation [5–9]. Rocks are mainly conglomerate in this reservoir with reserves of more than 1 billion tons and a huge development potential. Previous studies have shown that the conglomerate porosity and permeability is less than 10% and 5 mD, respectively [10]. Therefore, hydraulic fracturing is needed to improve the reservoir permeability, and it is necessary to grasp its mechanical properties.

Conglomerate is a kind of sedimentary rock which is widely distributed all over the world. Yet, little research has been conducted towards its mechanical properties due to its lower economic value. Liu *et al.* proposed a method to characterize rock heterogeneity and established its discrete element model based on numerical simulation software PFC2D. It indicates that uniaxial compressive strength (UCS) and tensile strength present a linear decreasing as the stress concentration increases due to the increase in heterogeneity [11]. Khanlari *et al.* studied the relationship between gravel content, contact pattern and the conglomerate mechanical properties. It shows that the conglomerate mechanical properties and petrological characteristics are highly correlated, in particular, the gravel percentage is closely related to the mechanical parameters [12]. Li *et al.* simulated the hydraulic fracturing propagation in the conglomerate based on discrete element. It shows that there are four modes between gravels in the conglomerate and the cracks: crack arrest, bypassing gravels, penetrating gravels and adsorbed by gravels [13]. Currently, no studies have been conducted on the correlation between conglomerate mechanical properties and gravel petrological characteristics, contact pattern and cementation conditions.

In addition, there were few studies on influences of particle size on mechanical properties of rocks. Olsson [14] studied and calculated the yield stress of marble with a particle size of 0.005–3.5 mm, and believed that the yield stress was negatively correlated with particle size in a nonlinear manner. Conglomerate with the particle size up to 30 mm also showed a similar rule. The larger the gravel particle size, the stronger the heterogeneity of particle size, the lower the compressive strength and Young's modulus and the more obvious the plastic characteristics of stress–strain curve [15–18]. Conglomerate is generally regarded as a two-phase composite material due to the great difference in mechanical properties between gravel and matrix. Research on gypsum and concrete also conformed to the characteristics of two-phase composite materials. Uniaxial compression results of artificial composite samples showed that the compressive strength is negatively correlated with the content of hard particles, and the stress concentration at the edge of hard particles led to cracking around hard particles [19,20]. The ice–rock mixture is also a two-phase combination, and a great deal of research has been carried out to study the mechanical properties of it in ice sheets or glaciers. When the content of rock particles was small (about 6–15% according to different studies), the ice–rock mixture behaved as pure ice, and became hardened and stronger as the proportion of mineral particles increased. When the content increased further (greater than or equal to 56%), the strength of the mixed phase was close to that of dry sand [21–23]. The law was similar to the strengthening effect of adding granular materials in metal materials [24–27]. Qi *et al.* [28] believed that the hard particles blocked the lattice dislocation and increased the overall strength of the composite phase. This was not consistent with the laws of rock, gypsum and concrete. All studies above have not systematically studied influences of gravel particle size and volume fraction on the brittle-plasticity and mechanical properties of rocks, which will be discussed in this paper.

As to the mechanical properties of the conglomerate in Baikouquan Formation and Urho Formation in Mahu Sag, we established the conglomerate numerical models and conducted uniaxial compression tests to obtain the stress–strain curve, compressive strength, elasticity modulus and the micro-crack propagation based on the particle flow numerical simulation software PFC2D. The influence of gravel content, geometry and cementation strength on mechanical properties of the conglomerate was analysed based on this research to provide theory foundation for hydraulic fracturing.

# 2. Particle flow code

Based on the discrete element methods, Cundall & Strack [29] proposed the particle flow code (PFC). Particles and bonds are used to characterize the rock material in PFC, which is called the bonded

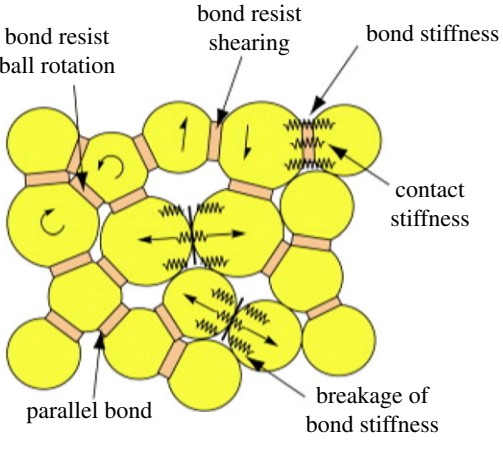

**Figure 1.** Schematic diagram of the PBM and its micro-mechanical behaviours.

particle model (BPM). Discs (PFC2D) or balls (PFC3D) of unit-thickness are used to represent discrete particles, which are considered rigid and have normal and shear stiffness. In this study, we built a uniaxial compression model of the conglomerate sample based on the parallel bond model (PBM), as shown in figure 1 [30]. PBM is mainly used to simulate the mechanical behaviour which is resistant to both force and moment, and can be considered as a set of springs that are parallel to the contact plane. Relative motion between particles could produce forces and torques at the parallel bond positions, which are strongly related to the maximum shear and normal stress around the contact plane. Once the maximum principal stress is higher than the corresponding bond strength, the parallel bond will be broken. Crack will be generated between the particles; the bond, force and torque at this position will then be removed. Under an external loading, the bond continues to break, and the total number of cracks gradually increases, which is the cumulative crack.

Specifically, the movement between particles is determined by Newton's second law, while the interaction between particles is determined by the force–displacement law. The force–displacement law of force and moment in the PBM can be illustrated as follows:

1. Update properties on the bond cross-section

$$\bar{R} = \bar{\lambda} \begin{cases} \min(R^{(1)}, R^{(2)}), & \text{ball} - \text{ball} \\ R^{(1)}, & \text{ball} - \text{facet} \end{cases}' \tag{2.1}$$

$$\bar{A} = \begin{cases} 2\bar{R}t, & \text{2D}(t=1) \\ \pi\bar{R}^2, & \text{3D} \end{cases}, \tag{2.2}$$

$$\bar{I} = \begin{cases} \dfrac{2}{3}t\bar{R}^3, & \text{2D}(t=1) \\ \dfrac{1}{4}\pi\bar{R}^4, & \text{3D} \end{cases} \tag{2.3}$$

and
$$\bar{J} = \begin{cases} 0, & \text{2D} \\ \dfrac{1}{2}\pi\bar{R}^4, & \text{3D} \end{cases}' \tag{2.4}$$

where $\bar{\lambda}$ is the radius multiplier, which is 1.0 by default. $\bar{A}$ is the parallel bond cross-sectional area. $\bar{I}$ is the inertia moment on the bond cross-section. $\bar{J}$ is the polar inertia moment on the bond cross-section.

2. Update the parallel bond force

$$\vec{\bar{F}} = \vec{\bar{F}}_n + \vec{\bar{F}}_s, \tag{2.5}$$

$$\bar{F}_n := \bar{F}_n + \bar{k}_n\bar{A}\Delta\delta_n \tag{2.6}$$

and
$$\bar{F}_s := \bar{F}_s + \bar{k}_s\bar{A}\Delta\delta_s. \tag{2.7}$$

As shown in formula (2.5), the parallel bond force between particles is decomposed into a normal force and a shear force. Equations (2.6) and (2.7) are the updated parallel bond force in each direction. $\bar{k}_n$ and $\bar{k}_s$

are the normal and shear stiffness, respectively. $\Delta\delta_n$ and $\Delta\delta_s$ are the relative increment of normal-displacement and shear-displacement, respectively.

3. Update the parallel bond moment

$$\bar{M} = \bar{M}_t\hat{n}_c + \bar{M}_b\,(\text{2D model}: \bar{M}_t \equiv 0), \tag{2.8}$$

$$\bar{M}_t := \begin{cases} 0, & \text{2D} \\ \bar{M}_t - \bar{k}_s\bar{J}\Delta\theta_t, & \text{3D} \end{cases} \tag{2.9}$$

and

$$\bar{M}_b := \bar{M}_b - \bar{k}_n\bar{I}\Delta\theta_b. \tag{2.10}$$

As shown in formula (2.8), the parallel bond moment is decomposed into a torque and a bending moment. $\hat{n}_c$ is the unit direction vector. Equations (2.9) and (2.10) are the updated torque and bending moment. $\theta_t$ and $\theta_b$ are the relative increment of torsion angle and bend-rotation, respectively

$$\bar{\sigma} = \frac{\bar{F}_n}{\bar{A}} + \bar{\beta}\frac{\|\bar{M}_b\|}{\bar{I}} \tag{2.11}$$

$$\bar{\tau} = \frac{\|\bar{F}_s\|}{\bar{A}} + \begin{cases} 0, & \text{2D} \\ \bar{\beta}\dfrac{|\bar{M}_t|\bar{R}}{\bar{J}}, & \text{3D} \end{cases} \tag{2.12}$$

$$\text{with } \bar{\beta} \in [0,1], \tag{2.13}$$

where $\bar{\sigma}$ and $\bar{\tau}$ are the normal stress and shear stress at the parallel bond periphery, respectively. $\bar{\beta}$ is the moment-contribution factor, ranging from 0 to 1.

4. Bond failure criteria between particles

While the maximum normal stress between particles is greater than the tensile strength ($\bar{\sigma} > \bar{\sigma}_c$), tensile failure occurs. Shear strength $\bar{\tau}_c = c + \sigma\tan\bar{\phi}$. $\sigma = \bar{F}_n/\bar{A}$ is the average normal stress on the parallel bond cross-section. While the maximum shear stress between particles is greater than the shear strength ($\bar{\tau} > \bar{\tau}_c$), shear failure occurs.

# 3. Construction of numerical model

## 3.1. Model construction

We conducted this study on conglomerate and applied PBM contact model to establish its discrete element model and study its mechanical characteristics and laws, based on the PFC2D particle flow software platform combining its geometric and mechanical characteristics. Samples were collected at depth of 3388–3445 m from Well A, 131 block, Mahu sag Junggar Basin (figure 2). Conglomerate contains a large amount of gravels which differ greatly in size and shape, and the gravel diameter varies between 2 and 25 mm.

To make the model more accurate, taking figure 3a as example, we first unfolded the outer surface of the cylindrical sample, as shown in figure 3b, then selected a region of 50 × 100 mm represented by the black dotted box in figure 3b to pick the gravel shapes, sizes and distribution and obtain the digital images which were finally imported into the PFC2D software to establish the two-dimensional model, as shown in figure 3c. Based on the conglomerate as shown in figure 1, we established a series of different models. To build the conglomerate model with different gravel contents, we took figure 3c as the basic model and removed some of the particles to obtain the conglomerate model with gravel volume contents ranging from 20 to 35% (area per cent in the model). Meanwhile, we established the conglomerate models with 55 and 75% gravel volume contents based on figure 2d,e, as shown in figure 3d.

The model generated a total of 11 346 disc particles with a radius ranging from 0.3 to 0.4 mm and the initial model porosity was 16%. Considering the gravel edge fracturing in the conglomerate, the harder gravels were simulated by flexible clusters which were bonded together by balls with certain strength. When the external force is sufficiently large, this bond could be broken. The generated clusters were divided into different groups by which contact types within the clusters, between the clusters and between the clusters and matrix particles were given so as to simulate the conglomerate [29]. Contour of the gravels was drawn based on the geometric model in figure 3 and particles within the gravel contour were circled to form the clusters. (As shown in figure 4, there are 26 typical clusters in this model.) Therefore, the conglomerate model consisted of three parts: gravels, matrix and the bond

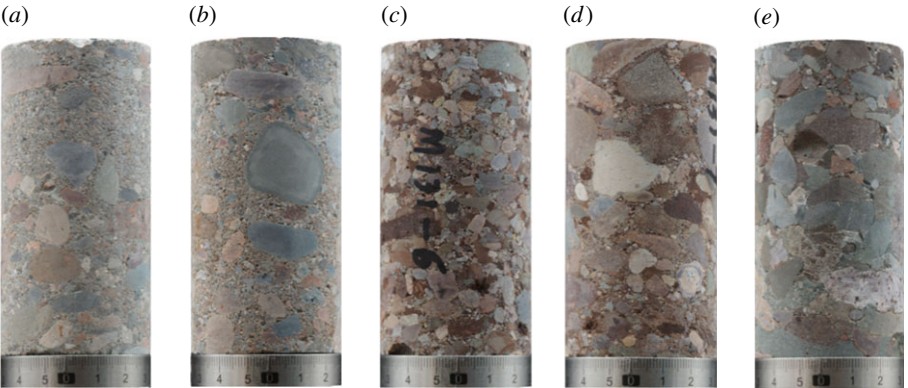

**Figure 2.** Conglomerate samples (sample size $\Phi 50 \times 100$ mm).

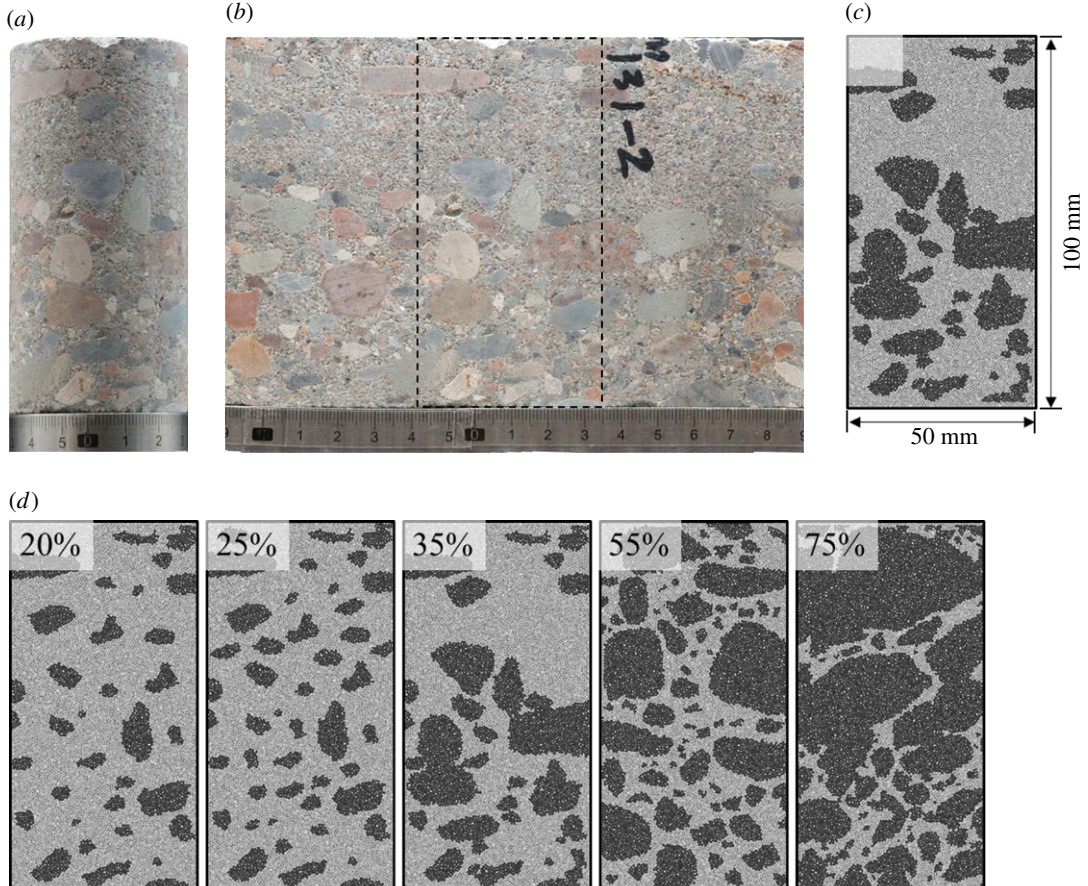

**Figure 3.** Numerical models of the conglomerate that contain different gravel contents (the black area represents gravels and the grey area represents matrix).

between gravels and matrix; micro-parameters of these three parts have to be constructed in order to study the conglomerate mechanical properties.

## 3.2. Model parameter calibration

Since there is no specific correspondence between the micro-parameters of particles in PFC and the macro-mechanical properties of the rocks, we adopted the method of trial and error to change the micro-parameters repeatedly till they met with the macro-mechanical properties of the material. In the conglomerate sample as shown in figure 2, the gravel is mainly andesite, and matrix is formed from the compaction and cementation of minerals such as quartz, feldspar, calcite and clay, while the

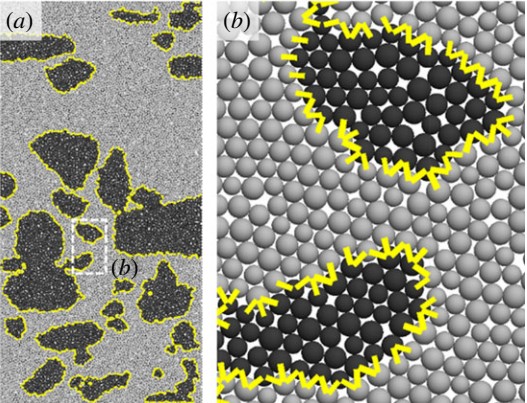

**Figure 4.** Numerical model of the conglomerate (the black spheres represent particles inside the gravels, the grey spheres represent the matrix particles, the yellow short lines represent the bond between gravels and matrix).

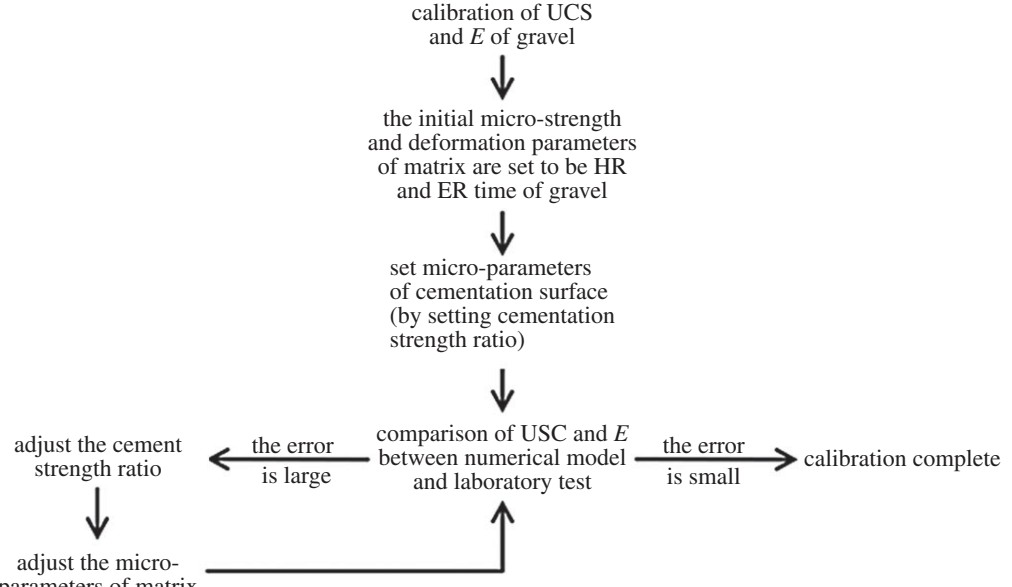

**Figure 5.** Flow chart of parameter calibration for conglomerate numerical model.

cementation between gravel and matrix is mainly argillaceous cementing, the average strength from high to low of which is: gravel > matrix > cementing between gravel and matrix. To simplify the numerical model, we assume that the gravel and the matrix are a single substance, and the macro-mechanical parameters of andesite are used to calibrate the micro-mechanical parameters of the gravel. The UCS of andesite is 175 MPa, while the elastic modulus ($E$) is 69.98 GPa.

Based on the conglomerate sample as shown in figure 2a, we established the numerical sample (figure 4), and obtained the average hardness and average elastic modulus of the gravel and matrix according to the micro-indentation. The calculated average hardness ratio (HR) of matrix to gravel is 0.08, and the average elastic modulus ratio (ER) of matrix to gravel is 0.27. Finally, we calibrated the conglomerate micro-mechanical parameters through the uniaxial compression test. Specific calibration flow chart is shown in figure 5.

Through repeated trial and error, the micro-mechanical parameters of gravel, matrix and cementing surface are determined, as shown in tables 1 and 2, respectively (the cementation strength ratio is set to 0.2). Figure 6 shows the uniaxial compressive stress–strain and cumulative crack number curves of gravel and matrix. The final calibration results are as follows: the UCS of gravel is 175 MPa, the elastic modulus is 69.98 GPa; the UCS of the matrix sample is 110 MPa, while the elastic modulus is 13.19 GPa. Comparison between the mechanical parameters gained through the conglomerate numerical model and laboratory test is shown in table 3. The errors of compressive strength and elastic modulus are

**Table 1.** Parameters of micro-mechanical properties of the matrix and gravels.

| | matrix | gravels |
|---|---|---|
| minimum particle size (mm$^{-1}$) | 0.3 | 0.3 |
| particle size ratio (maximum/minimum) | 1.33 | 1.33 |
| particle density (kgm$^{-3}$) | 2500 | 2875 |
| friction coefficient between particles | 0.8 | 0.8 |
| particle contact Young's modulus (GPa) | 10.0 | 45.0 |
| particle contact normal to shear stiffness ratio | 4.0 | 2.0 |
| parallel bond cohesion (MPa) | 50.0 | 80.0 |
| parallel bond normal bond strength (MPa) | 50.0 | 80.0 |
| parallel bond frictional angle (°) | 35.0 | 35.0 |
| parallel bond Young's modulus (GPa) | 10 | 45 |
| parallel bond normal to shear stiffness ratio | 4.0 | 2.0 |

**Table 2.** Micro-mechanical parameters of the cementing surface.

| cementation strength ratio[a] | parallel bond Young's modulus (GPa) | parallel bond normal to shear stiffness ratio | parallel bond normal bond strength (MPa) | parallel bond cohesion (MPa) | friction coefficient between particles |
|---|---|---|---|---|---|
| 0.1 | 4.5 | 0.2 | 8.0 | 8.0 | 0.08 |
| 0.2 | 9.0 | 0.4 | 16.0 | 16.0 | 0.16 |
| 0.3 | 13.5 | 0.6 | 24.0 | 24.0 | 0.24 |
| 0.4 | 18.0 | 0.8 | 32.0 | 32.0 | 0.32 |
| 0.5 | 22.5 | 1.0 | 40.0 | 40.0 | 0.40 |
| 0.6 | 27.0 | 1.2 | 48.0 | 48.0 | 0.48 |
| 0.7 | 31.5 | 1.4 | 56.0 | 56.0 | 0.56 |
| 0.8 | 36.0 | 1.6 | 64.0 | 64.0 | 0.64 |
| 0.9 | 40.5 | 1.8 | 72.0 | 72.0 | 0.72 |
| 1.0 | 45.0 | 2.0 | 80.0 | 80.0 | 0.80 |

[a]Cementation strength ratio refers to the ratio of micro-mechanical parameters of the cementing surface to those of the gravels. In the parallel contrast experiment, micro-mechanical parameters of the matrix and gravels remained the same, only the cementing surface parameters were adjusted. And Young's modulus and normal to shear stiffness ratio of the particle contact and bond contact are the same.

9.42% and 4.93%, respectively. The final failure characteristics (figure 7) are in good agreement with the test results (figure 7b). The red lines represent tensile cracks, while the green lines represent shear cracks. Therefore, the selection of micro-mechanical parameters is reasonable.

## 3.3. Research process

The gravel size, content and component, etc., differ greatly in conglomerate due to the differences of sedimentary environment in Mahu. Therefore, many factors could affect the mechanical properties of the conglomerate. In this study, effects of cementation strength, gravel content and shape on the mechanical parameters and crack law of the conglomerate were mainly studied.

Common cementing types are argillaceous cementing, calcareous cementing and a little siliceous cementing, and the difference among these types could lead to different cementation strength. In this paper, different cementing strength between the matrix and gravels was set to study the effect of

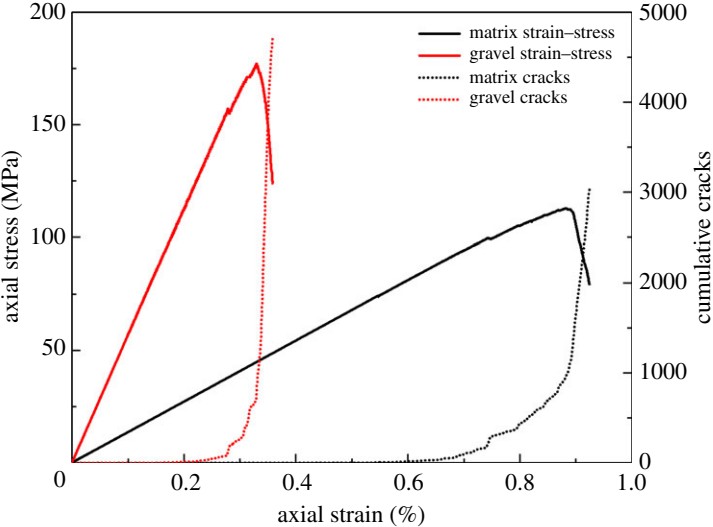

**Figure 6.** Uniaxial compression stress–stress curves and cumulative cracks curves of gravels and matrix.

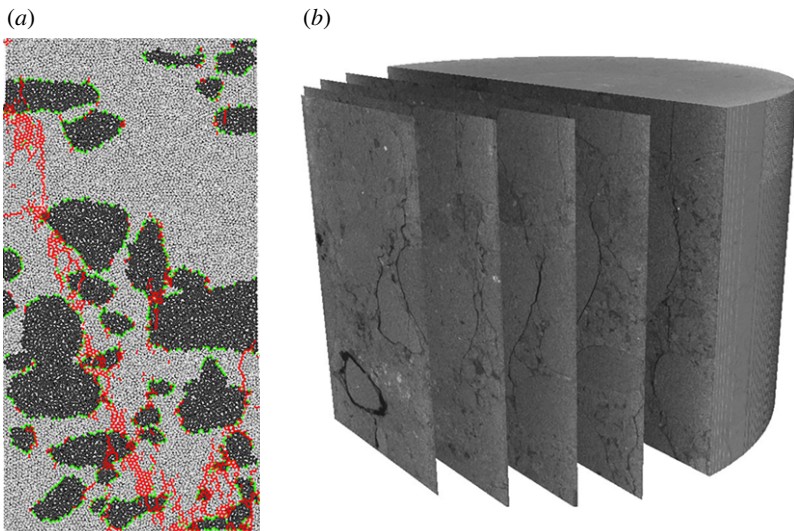

**Figure 7.** Failure mode of complete conglomerate sample gained from (*a*) numerical simulation and (*b*) experiment.

**Table 3.** Comparison of mechanical parameters between numerical model and laboratory test.

| parameter | numerical model | test sample | Error (%) |
|-----------|-----------------|-------------|-----------|
| UCS (MPa) | 66.2 | 60.5 | 9.42 |
| $E$ (GPa) | 19.17 | 18.27 | 4.93 |

cementing strength on the mechanical properties. When the conglomerate is under an external force, the strength of the cementing surface greatly affects the development of internal cracks. In §3.2, reasonable macro-mechanical parameters were defined by calibrating the micro-parameters. Yet, there is still no reasonable approach to directly measure the mechanical parameters of the cementing surface between matrix and gravels. To study the effect of cementing strength on the mechanical properties of conglomerate, we set the deformation parameters (parallel bond Young's modulus, parallel bond normal to shear stiffness ratio), strength parameters (parallel bond normal bond strength, parallel bond cohesion) and friction coefficient to 0.1–1.0 times the gravel micro-parameters, respectively (table 2), while keeping the other parameters unchanged. The multiple relation is called cementation strength

ratio. Then, 10 groups of uniaxial compression tests were conducted. Due to the difference between macro- and micro-coefficients, the macro-mechanical parameters of the cementing surface will not be 0.1–1.0 times those of the gravels. Therefore, in this section, we only studied the effect of cementing surface strength on the macro-mechanical properties of conglomerate from micro-perspective.

In this section, we used the numerical model of conglomerate with a gravel content of 35%. During the uniaxial compression test, load was applied through applying displacements on the top and bottom walls and the displacement rate applied by the device should be sufficiently slow to ensure the loading is quasi-static. The loading rate in this study was $0.025 \text{ m s}^{-1}$, and the sample lost stability gradually and cracked as the axial pressure increased. Tests conducted in §4.1 and 4.2 were both stopped when the axial stress was 60% of the peak strength, while to speed up the progress, the test conducted in §4.3 was stopped when the axial stress was 70% of the peak strength.

To study the effect of different gravel contents, five models as shown in figure 3d were used to conduct the uniaxial compression tests with gravel content of 20%, 25%, 35%, 55% and 75%, respectively (figure 3). A parameter package in which the cementation strength ratio was 0.2 (table 2) was used as the micro-parameters of the cementing surface. Meanwhile, since the models in figure 3d were derived from actual samples, to avoid the instability caused by the irregular shapes, we used spherical particle to study the effect of gravel content in §4.3. The spherical gravel content was 5–75% with a radius of 5–5.5 mm, and details of the micro-mechanical parameters are shown in table 1, the models are shown in figure 19.

# 4. Results

## 4.1. Effects of cementation strength on the mechanical characteristics

Figure 8 shows the uniaxial compression stress–strain results when the cementation strength ratio is 0.1–1.0, which indicates that the compressive strength and elasticity modulus decrease significantly with the cementation strength ratio decreasing from 1.0 to 0.1. The post-peak drops also show the tendency from brittle failure to plastic failure. From the number of cumulative cracks, we can find that the smaller the cementation strength ratio is, the earlier the crack initiation occurs inside the rocks.

In order to analyse the effect of cementation strength ratio on the mechanical parameters, we collected and calculated the UCS, total number of cracks, crack initiation point strength/UCS and the elasticity modulus data, respectively, to seek the relationship between them. Figure 9 shows all the data, and the four parameters all change regularly with the increasing cementation strength ratio. Crack initiation point strength is the axial stress corresponding to the first crack occurred in the conglomerate numerical model in uniaxial compression test. Therefore, crack initiation point strength/ UCS is an important parameter to evaluate the initial crack time inside the rock. The total number of cracks is the number of all the cracks in the model after the uniaxial compression test, that is, the number of all the cracks in the corresponding crack shape map.

The UCS increases rapidly when the cementation strength ratio is less than 0.6 and tends to be stable thereafter. The total number of cracks, crack initiation point strength/UCS and the elasticity modulus each reaches its critical value while the cementation strength ratio is 0.3. When the cementation strength ratio is less than 0.3, the cracks develop much earlier and the total number is also much more. And the three parameters get stable while the cementation strength ratio is greater than 0.3, cracks cease to be generated and the elasticity modulus tends to be unchangeable.

The crack shapes also change greatly with the increase in mechanical parameters at the cementing surface. Figure 10 shows the crack shapes of the conglomerate samples with different cementation strength ratios after uniaxial compression. The fractal theory could reflect the effectiveness of the space that occupied by complex shapes; it is a measure of the irregularities of complex shapes. Therefore, in this paper, we used fractal dimension to quantitatively evaluate the complexity of crack morphology. The fractal dimension $D$ is given as follows:

$$D = \lim_{\varepsilon \to 0} [\log N(\varepsilon) / \log (1/\varepsilon)]. \tag{4.1}$$

In this paper, a small cube with side length $\varepsilon$ is used to cover the cracks, and $N(\varepsilon)$ is the number of small cubes required to cover the cracks. The specific calculation is done using the FracLab toolbox in Matlab. First, FracLab automatically generates the upper and lower limits of $\varepsilon$ according to the size of the picture. Then, choose how many small cubes of different sizes to generate. Different sizes of $\varepsilon$

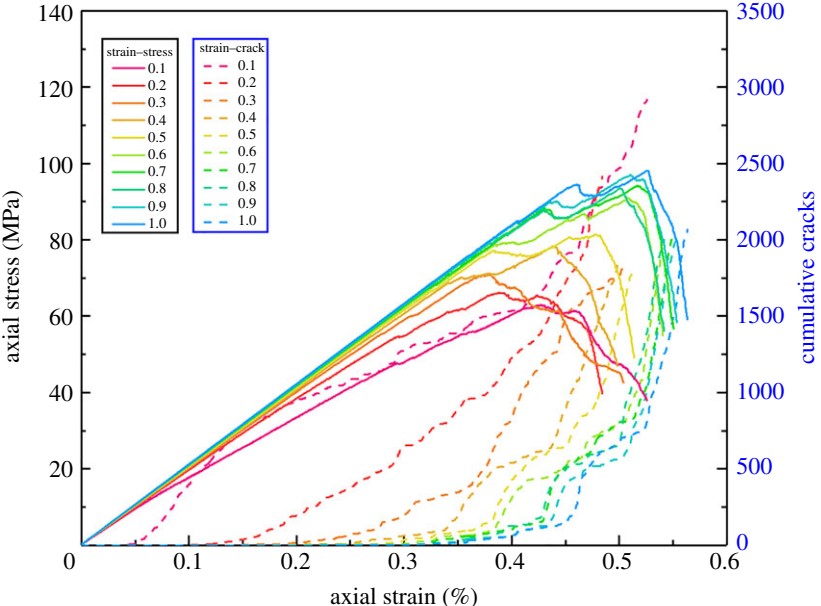

**Figure 8.** Uniaxial compression stress–strain and cumulative cracks curves of the conglomerate with different cementation strength ratios.

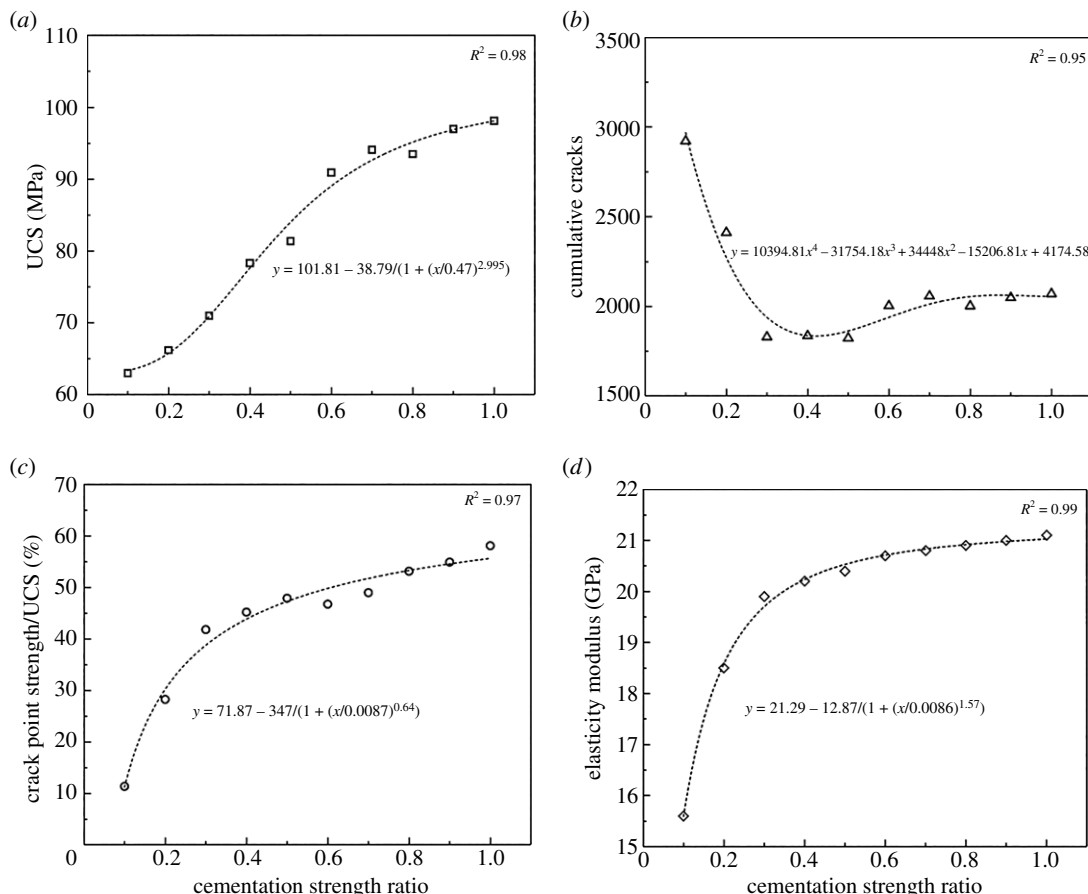

**Figure 9.** Effect of cementation strength ratio on mechanical parameters and cumulative cracks.

correspond to different $N(\varepsilon)$. The progression of $\varepsilon$ can be power law or linear. Finally, draw a scatter plot with x-axis $\log_2(\varepsilon)$ and y-axis $\log_2(N)$ in a two-dimensional coordinate system. Fit the scatter plot into a straight line, and the opposite number of the slope of the straight line is the fractal dimension $D$.

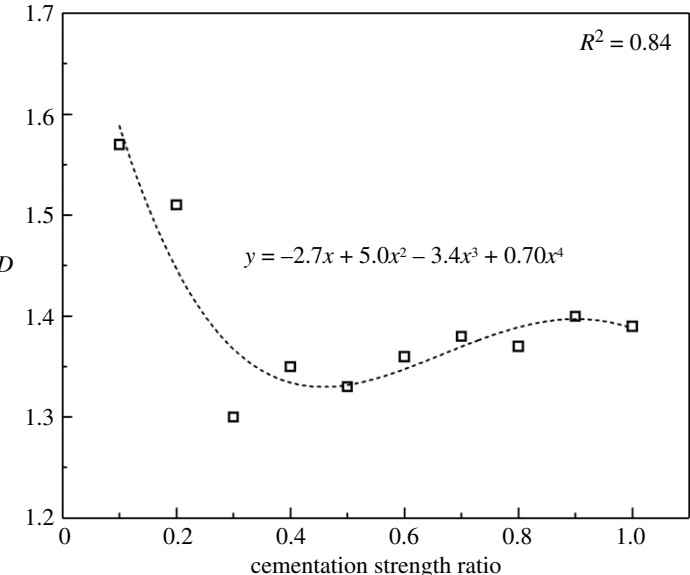

**Figure 10.** Crack shapes under different cementation strength ratios.

$$y = -2.7x + 5.0x^2 - 3.4x^3 + 0.70x^4$$

$R^2 = 0.84$

**Figure 11.** Relationship between cementation strength ratio and fractal dimension.

Combining figures 9*b* and 11, we can see when the cementation strength ratio is less than 0.3, the fractal dimension is much larger and plenty of annular shear cracks are generated bypassing gravels which form a complex crack network with tensile cracks. However, when the cementation strength

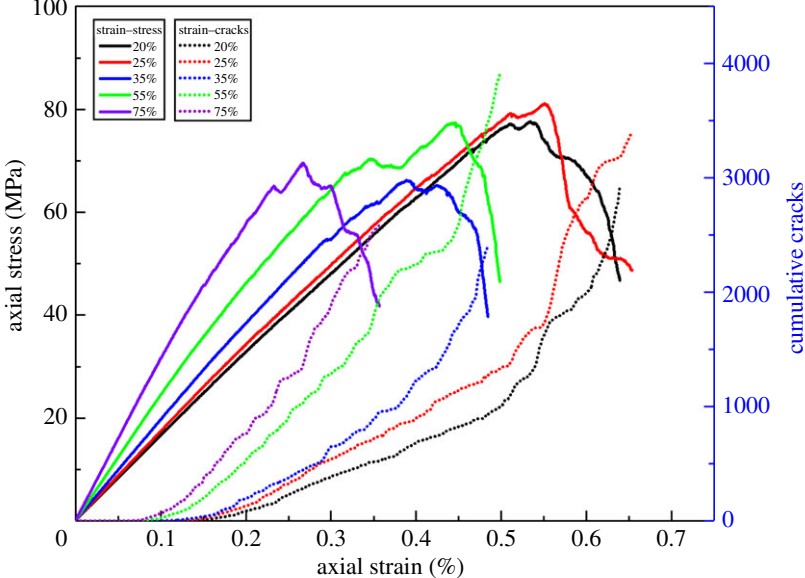

**Figure 12.** Strain–stress results of the conglomerate with different special-shaped gravel contents.

ratio is larger than 0.3, the fractal dimension is relatively smaller and tends to be stable, tensile cracks are in majority and crack shapes are relatively simpler on the whole. To sum up, the cementation strength between gravels and matrix in the conglomerate mainly influences the mechanical properties which are related to the initiation of micro-cracks and formation of shear cracks bypassing gravels. All the parameters reach their critical values when the cementation strength ratio is 0.3.

## 4.2. Effect of special-shaped gravels on the mechanical characteristics

Gravels with various shapes can be seen from the conglomerate sample pictures as shown in figure 2. Based on the actual shapes, we established the numerical models of special-shaped gravels as shown in figure 3$d$. Figure 12 shows the strain–stress results of the conglomerate model with different gravel contents. From the curves we can see, with the increasing gravel content, the UCS decreases while the elasticity modulus gets greater and cracks begin to occur much earlier.

To analyse the effect of special-shaped gravel contents on the mechanical parameters, we collected and calculated the UCS, total number of cracks, crack initiation point strength/UCS and the elasticity modulus data, respectively, to seek the relationship between them. Figure 13 shows the data; as the special-shaped gravel content increases, the crack initiation point strength/UCS and the elasticity modulus also increase and show regular linear changes. However, we did not find an obvious changing rule between the UCS, total number of cracks and the special-shaped gravel content. And the determining coefficients of linear fitting are all less than 0.2. These are all contributed to the various shapes of actual gravels and too few data points. Therefore, we conducted the tests specifically on spherical gravels in §4.3, and the models can be seen in figure 19.

Figures 13 and 14 show the corresponding crack propagation characteristics. Cracks were extracted separately so as to observe their shapes, as shown in figure 15. And the cracks exhibit three types of encounter behaviours in the gravels: bypassing gravels, penetrating directly and embedded in gravels. As shown in figures 14, 15 and 16, cracks mainly propagate bypassing the gravels when the gravel volume content is 20–35%, except for the shear cracks around gravels. While the gravel volume content is 75%, cracks that penetrate directly through the gravels are in majority. And while the gravel volume content is 55%, cracks bypassing and penetrating directly through the gravels both exist. Obviously, gravel content could dominate the crack propagation patterns.

## 4.3. Effect of spherical gravels on the mechanical characteristics

To eliminate the effect of gravel shape on the research of different gravel contents, ideal spherical gravels are proposed, as shown in figure 19. Figure 17 shows the uniaxial compression stress–strain curve results of the spherical gravel model with different gravel contents. While the gravel volume content is 5–75%, as

(a) 
$y = -11.84x + 79.35$

$R^2 = 0.19$

UCS (MPa)

(b) 
$y = -136.88x + 3107.69$

$R^2 = 0.003$

cumulative cracks

(c) 
$y = 13.9x + 18.49$

$R^2 = 0.70$

crack point strength/UCS (%)

gravel content (%)

(d) 
$y = 25.31x + 9.57$

$R^2 = 0.96$

elasticity modulus (GPa)

gravel content (%)

**Figure 13.** Effect of different special-shaped gravel contents on mechanical parameters and cumulative cracks.

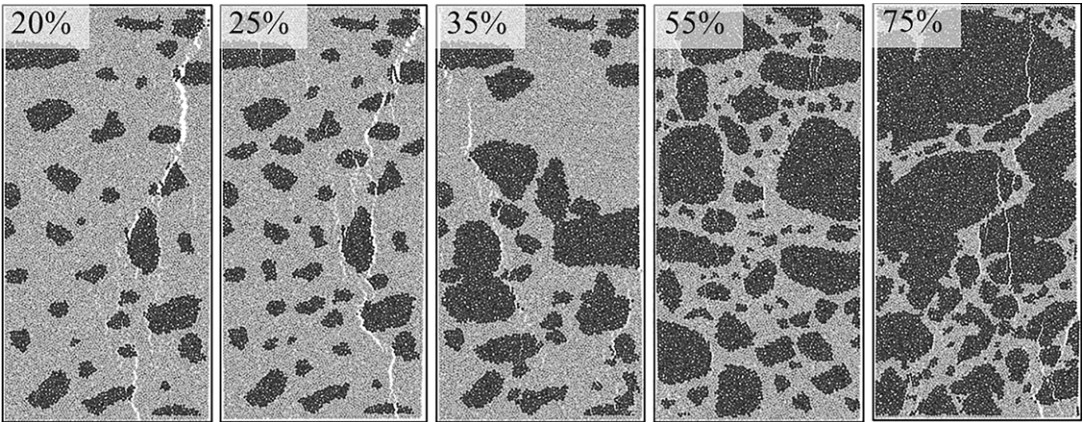

**Figure 14.** Crack shape after fracturing with different special-shaped gravel contents.

the spherical gravel content increases, the elasticity modulus also increases and the UCS first decreases then increases.

To analyse the effect of spherical gravel contents on the mechanical parameters, we collected and calculated the UCS, total number of cracks, crack initiation point strength/UCS and the elasticity modulus data separately to seek the relationship between them. As shown in figure 18, the four parameters all change regularly with the increasing spherical gravel content. The elasticity modulus increases, while the total number of cracks and crack initiation point strength/UCS first decrease then increase and decrease again, and the change of crack initiation time shows the same trend. Changes of the two parameters mainly occur while the spherical gravel content is 20–65%.

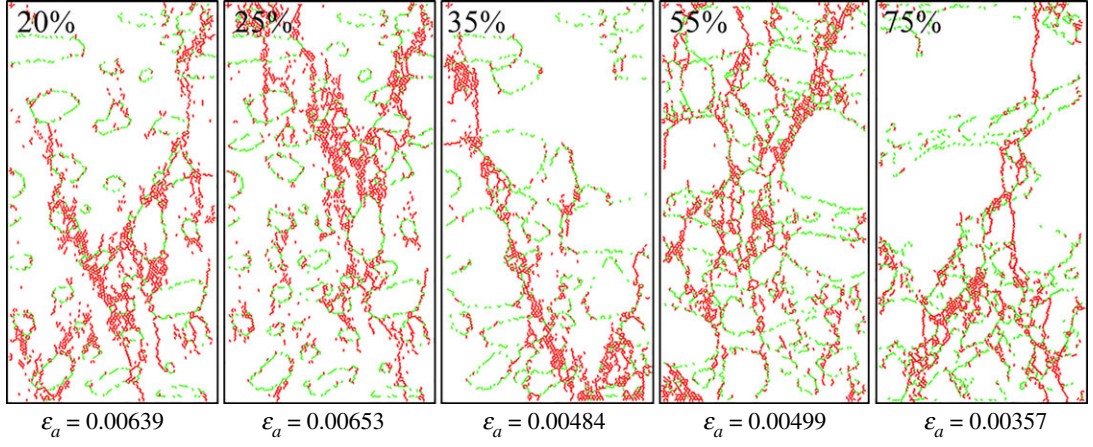

**Figure 15.** Crack shape under different special-shaped gravel contents.

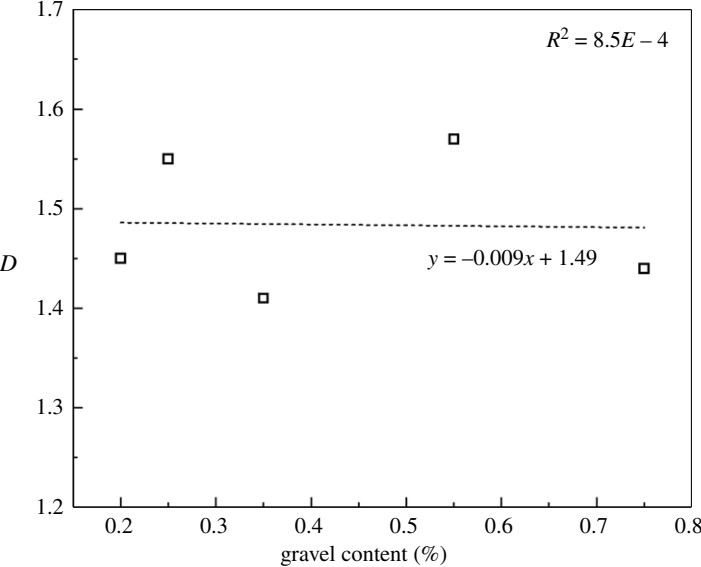

**Figure 16.** Relationship between special-shaped gravel content and crack fractal dimension.

Figure 19 shows the crack shapes after fracturing. Cracks were extracted separately so as to observe their shapes, as shown in figure 20. Tensile cracks are in majority while the gravel volume content is less than 10% and few cracks bypassing gravels are seen. As the gravel volume content increases, the number of shear cracks bypassing gravels also increases, and crack shapes get more complex. While the gravel volume content is 65% (as shown in figure 18*b*), the number of cumulative cracks reaches the peak, while the cracks become the most complex and the fractal dimension is the largest, as shown in figure 21. When the gravel volume content is larger than 65%, as the content increases, the number of cumulative cracks decreases, as shown in figure 18*b*, fractal dimension decreases, as shown in figure 21, while crack shapes get much simpler, the number of shear cracks bypassing gravels is reduced. Oblique crossing cracks are formed and most are tensile cracks.

# 5. Discussion

## 5.1. Analysis of the cracking characteristics of conglomerate

The mechanical properties of conglomerate are closely related to the three factors as discussed in §4. The conglomerate strength increases as the cementation strength increases. Shear cracks bypassing gravels tend to be formed more easily at a lower cementation strength and could form a complex crack network combined with tensile cracks. While, a higher cementation strength could generate oblique

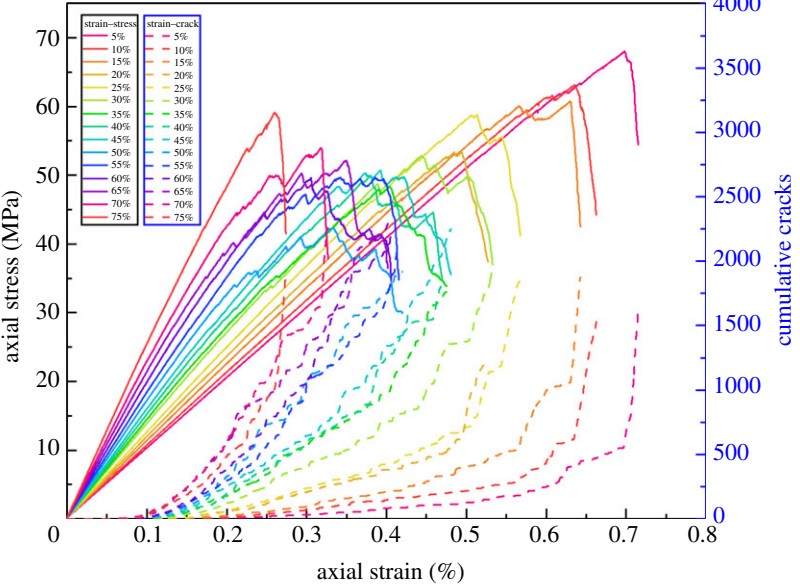

**Figure 17.** Stress–strain results of the spherical gravel model with different gravel contents.

*(a)*

$y = 97.43x^3 - 2.17x^2 - 69.42x + 70.64$

$R^2 = 0.87$

*(b)*

$y = 1783.35 - 30.15x + 1.34x^2 - 0.012x^3$

$R^2 = 0.63$

*(c)*

$y = -124.5x^3 + 168.1x^2 - 58x + 29.4$

$R^2 = 0.31$

*(d)*

$y = 20.9x^2 + 3.4x + 10.1$

$R^2 = 0.99$

**Figure 18.** Effect of different spherical gravel contents on the mechanical parameters and cumulative cracks.

crossing cracks which are mainly tensile cracks, and the crack shapes are relatively simpler. Results from the model based on actual samples and the ideal spherical model all indicate that, with the increasing gravel content, the conglomerate strength first decreases then increases. While the gravel volume

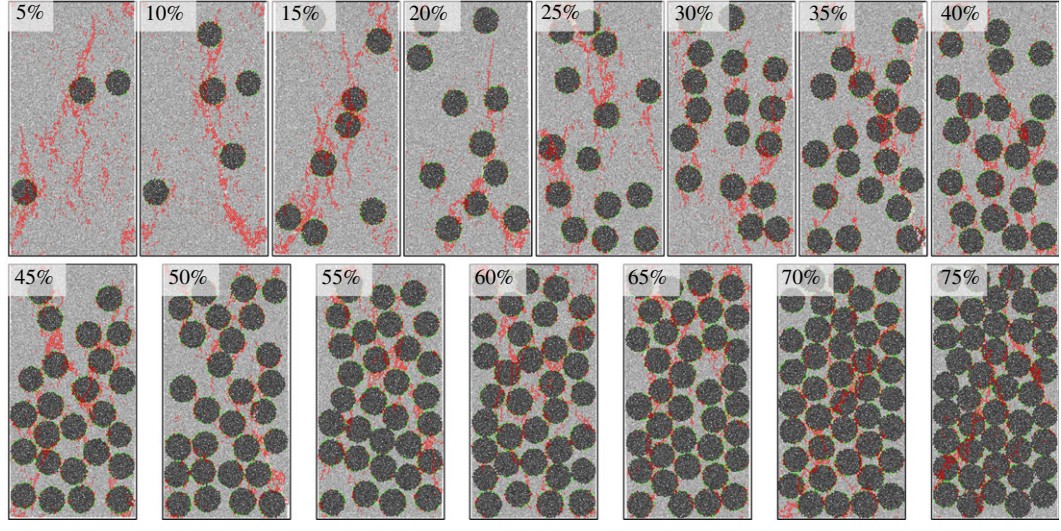

**Figure 19.** Crack distribution under different spherical gravel contents.

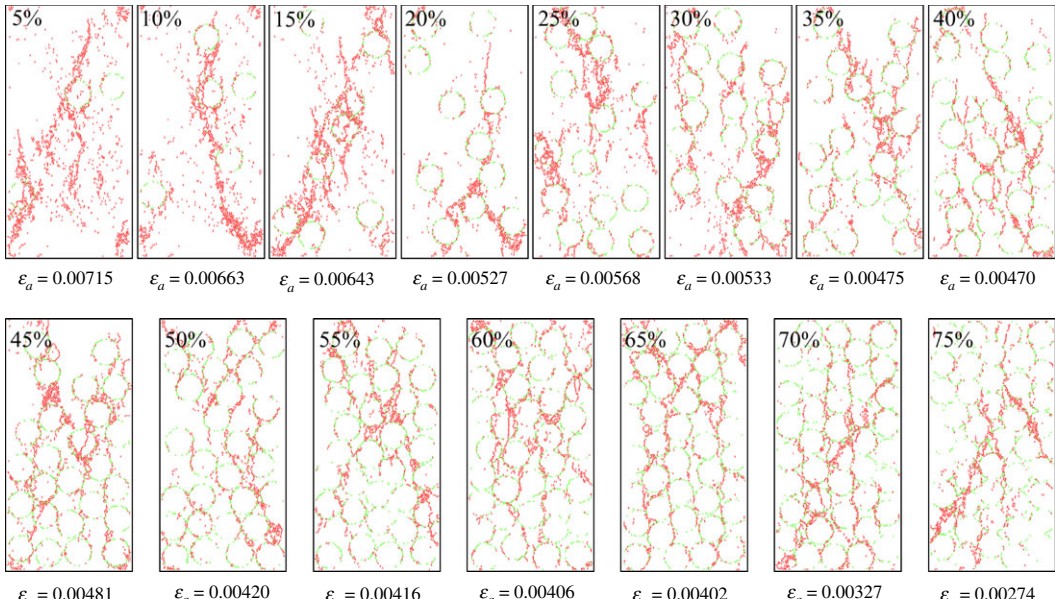

**Figure 20.** Crack shapes under different spherical gravel contents.

content is relatively lower (less than 10%) or higher (larger than 65%), a simple crack network would be generated. However, the formation mechanisms differ from each other, cracks most bypass the gravels when the gravel content is less than 10%, while when it is more than 65%, cracks penetrating gravels are in the majority. Crack network is very complicated while the gravel content is 10–65%; bypassing and penetrating gravels both dominate the formation of cracks.

The above findings indicate that the conglomerate cementation strength is relatively low while the cementation strength ratio is less than 0.3. And under the effect of deviatoric stress, additional stress is generated in the gravels due to the stress concentration, which further generates a shear stress on the gravel edge and forms shear cracks bypassing gravels. All these make local cracks more likely to develop and occur much earlier, and lead to a lower integral conglomerate strength. As the cementation strength ratio increases, the cementation strength between gravels and matrix also increases, yet the strain incompatibility gets weak, the additional stress caused by the stress concentration also decreases, which could make the rock more homogeneous and less easy to generate cracks bypassing gravels. As a result, the integral conglomerate strength gets greater, the cracks occur much later and total number of cracks gets less.

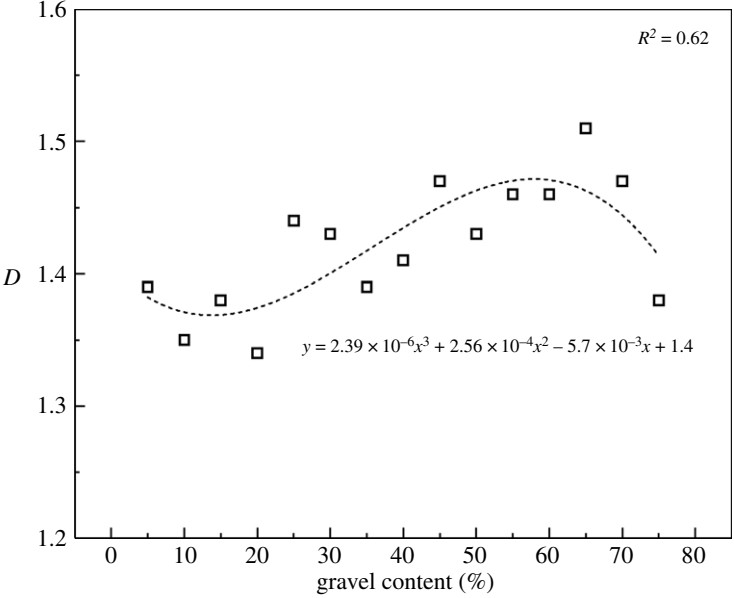

**Figure 21.** Relationship between the spherical gravel content and crack fractal dimension.

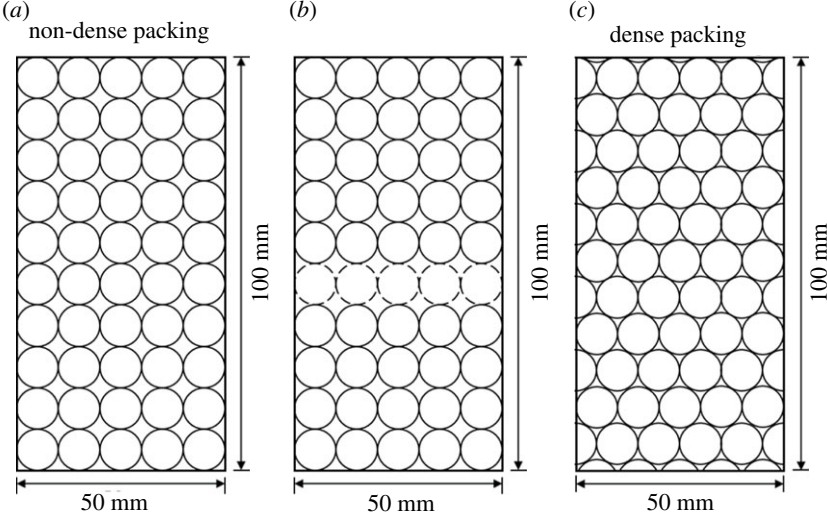

**Figure 22.** Two types of plane packing models with a gravel radius of 5 mm (the dotted lines represents the gravels that were removed). (*a*) 78.5%, (*b*) 70.7% and (*c*) 88.6%.

## 5.2. Micro-mechanism that mainly influences the conglomerate crack characteristics

The research findings show that gravel content has a great effect on the mechanical properties and crack characteristics. This effect is due to the existence of gravels as heterogeneous bodies, and the stress concentration generated from which is related to the distance between gravels. Under the condition of the same particle size, with higher gravel content and much closer distance between gravels, the additional stress generated is relatively much larger. Meanwhile, the rock bridge to be broken between two gravels is much shorter and more prone to crack. Therefore, as the gravel content increases, the conglomerate strength decreases and more cracks bypassing gravels are developed. However, if the gravel content is so high that the gravels contact with each other, the cementation generated would be a type between gravels rather than the weak one between gravels and matrix. Thus, while the gravel content is too high, even if the strength increases, cracks bypassing gravels would not develop.

Affected by the gravel content, while the gravel content is too high, the gravels contact between each other and the conglomerate sample is supported by gravels instead of matrix. The change of the support pattern leads to the changes of mechanical properties and crack shapes. The support pattern changes while the gravel content is 65%, based on the discussions in §5.1. The gravel critical content ($G_{CC}$) corresponding to

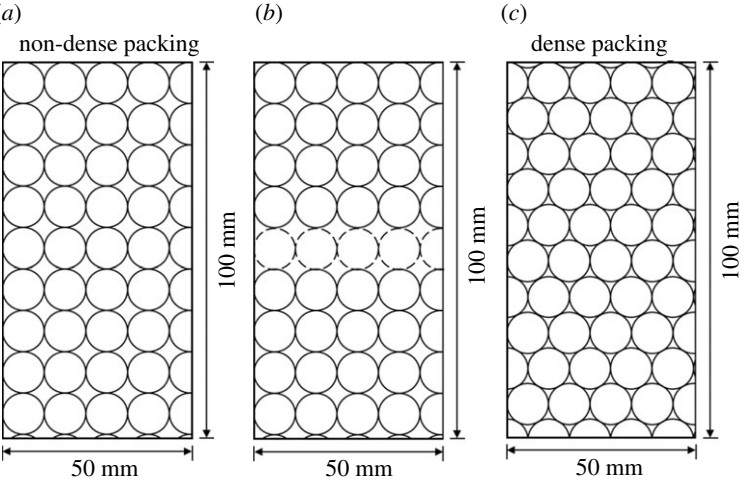

**Figure 23.** Two types of plane packing models with a gravel radius of 5.5 mm (the dotted lines represents the gravels that were removed). (*a*) 77.9%, (*b*) 69.3% and (*c*) 89.7%.

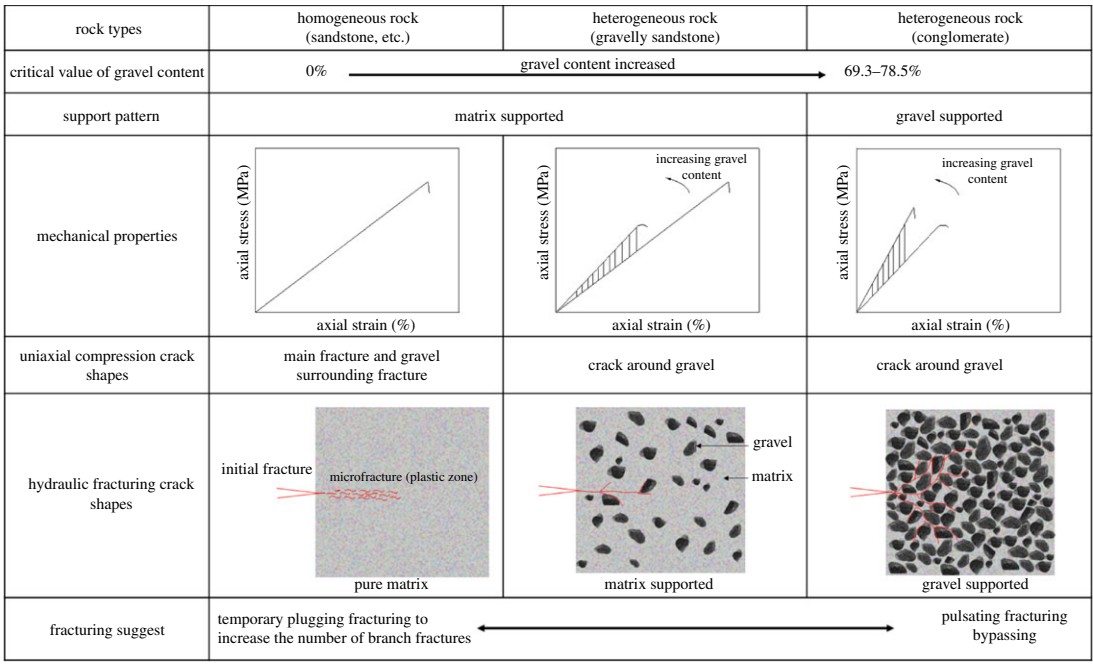

**Figure 24.** Countermeasure analysis chart of the conglomerate hydraulic fracturing.

the changing point of theoretical support pattern was calculated through plane packing model. The gravel contents corresponding to non-dense packing and dense packing at $R = 5$ mm and $R = 5.5$ mm were calculated, respectively, as shown in figures 22 and 23. While the packing is non-dense, the support pattern is already gravel supported, and the gravel contents are 78.5% and 77.9%, respectively. A strong force chain is formed throughout the gravels under uniaxial compression. Yet, it was not formed in the entire sample when the conglomerate is supported by matrix. Thus, to get the $G_{cc}$ corresponding to the changing point of support pattern, the dotted spherical gravels were removed from the model (figures 22*b* and 23*b*). The calculated gravel contents at $R = 5$ mm and $R = 5.5$ mm are 70.7% and 69.3%, respectively. To sum up, while the spherical gravel radius is at 5–5.5 mm, the $G_{cc}$ calculated are from 69.3% to 78.5%.

## 5.3. Enlightenment of the crack characteristics of conglomerate on hydraulic fracturing

Based on the above analysis of the conglomerate mechanical properties and crack shapes, we find that gravel content is the major control factor. And there are two types of conglomerates according to the

gravel content: matrix supported and gravel supported as shown in figure 24. Dominated by the gravel content, the local stress concentration in the conglomerate gets more obvious and the weak cementing surface between gravels and matrix tends to crack easily and generate cracks bypassing gravels. Therefore, the more cracks bypassing gravels there are, the more complex the cracks generated from hydraulic fracturing would be. Figure 24 shows that crack shapes of homogeneous rocks (e.g. sandstone) are relatively simple; therefore, temporary plugging is needed to force the opening of branch cracks to improve the crack complexity. As to the conglomerate with a high gravel content, cracks bypassing gravels are one of the benefits. Cyclic loading is applied on the cementing surface between gravels and matrix through pulsating fracturing to open cracks bypassing gravels at a lower fracturing pressure and finally form a complex crack network which consists of cracks bypassing gravels and principal cracks.

# 6. Conclusion

A discrete element numerical model of the conglomerate was established in this paper based on PFC2D to study the influence law of cementation strength between gravels and matrix, gravel content and shape on the mechanical properties and crack characteristics. Main conclusions can be drawn as follows:

(1) As the cementation strength decreases, the compressive strength decreases while the elasticity modulus increases clearly. The post-peak drop also shows the tendency of changing from brittle crack to plastic crack, and cracks change from tensile to shear cracks bypassing gravels, crack shapes get more complex and the critical value of cementation strength ratio is 0.3.

(2) While the gravel content is less than 10% and with its increasing, the conglomerate strength decreases and cracks bypassing gravels form a simple network. While the gravel content is 20–65% and as it increases, the conglomerate strength also decreases, yet cracks bypassing gravels form a complex network. And while the gravel content is greater than 65%, the conglomerate strength increases with its increasing, cracks penetrating gravels are in majority and form a simple crack network.

(3) A lower cementation strength and higher gravel content would lead to the increasing conglomerate heterogeneity, which is basically due to the change of support pattern. The gravel critical content of matrix-supported and gravel-supported rocks is between 69.3 and 78.5%, based on the equal-diameter particle packing model. While the gravel content is less than this value and with its increasing, the UCS decreases and cracks get more complex. Yet, while the gravel content is larger than this critical value, the above two parameters change in the opposite.

(4) In the homogeneous sandstone reservoir, temporary plugging and hydraulic fracturing are adopted to force the opening of branch cracks to increase the crack network complexity. While in conglomerate reservoir, cracks bypassing gravels are generated through pulsating fracturing with a lower fracturing pressure to form a complex crack network.

Data accessibility. Data available from https://doi.org/10.5281/zenodo.4451866.

Authors' contributions. H.G. proposed research ideas and ideas, and provided research funding support. S.L., W.Z. and J.W. carried out the concrete research work of experiment and numerical simulation. Y.S., P.L. and J.L. participated in the revision of the paper.

Competing interests. We declare that we do not have any commercial or associative interest that represents a conflict of interest in connection with the work submitted.

Funding. This work was supported by the Natural Science Youth Project of university scientific research plan in Xinjiang (XJEDU2021Y053), the Talent introduction research project of China University of Petroleum—Beijing at Karamay (XQSQ20200056), Development of Conglomerate Reservoir Laboratory in Xinjiang (grant no. 2019D04008), the Strategic Cooperation Technology Projects of CNPC and CUPB (grant no. ZLZX2020-01).

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
