## [Peer Review File · Royal Society Open Science]

Review History

RSOS-202178.R0 (Original submission)

Review form: Reviewer 1

Is the manuscript scientifically sound in its present form?

Yes

Are the interpretations and conclusions justified by the results?

Yes

Is the language acceptable?

Yes

Do you have any ethical concerns with this paper?

No

Have you any concerns about statistical analyses in this paper?

No

Recommendation?

Major revision is needed (please make suggestions in comments)

Comments to the Author(s)

The technical article investigates influence of cementation strength between gravels and matrix, gravel content and shape on the mechanical properties and crack characteristics of conglomerate by applying discrete element based numerical simulation. The manuscript is as such well written and includes interesting observations. However, the reviewer strongly advises to address the following points in a revised submission:

1. Instead of mentioning just the DEM software package, the main governing equations involved in DEM should be presented in Section 2.
2. The contact model employed, i.e. the Parallel Bond Model should be discussed in details with related mathematical equations. Further, the interaction between gravel-gravel particle, matrix-matrix particle and gravel-matrix particle should be addressed with reference to the applied bond model. In Section 3.1 (line no. 117), it should be clearly mentioned how many numbers of clusters were considered to model a typical gravel particle.
3. The procedure of micro-mechanical parameter calibration (Section 3.2) is not very clear. In one place, it has been stated that "In this study, we didn't correspond the conglomerate model with the laboratory experiment and the micro parameters of matrix and gravels were calibrated respectively" and in other place, it is mentioned that "The micro parameters were finally set to those corresponded to the cementation strength ratio of 0.2 (see in Table 2) by the method of trial and error in which the micro parameters were changed repeatedly. It should be clearly mentioned that the micro-parameters of agglomerates are calibrated against which macro-level data? Further, deviation observed between the experimental result and numerical simulation referred in Fig. 6 should be clearly explained.
4. It is not clear how the crack and cumulative crack is getting defined in the conducted simulations.
5. The cumulative cracks referred in Fig. 9(b) are at which strain level? Similarly, the strain level for the crack patterns shown in Fig. 10, 14, 19 and 20 should be also mentioned.
6. What is crack point strength?
7. At what strain level the elastic modulus referred in Fig. 9(d) has been measured? The initial elastic tangent modulus values for different strength ratio as shown in Fig. 8 seem very similar.
8. What is the definition of the term fractal dimension and how is it getting calculated in Fig. 11?
9. It is not clear how various special shapes gravels are getting characterized in the DEM simulations in comparison to the spherical shaped one. The lack of such shape characterisation might have resulted into high scatter in the results reported in Fig. 13(a-c). The same might be the reason behind not observing any distinct trend in Figure 16.
10. Figure 14 and 15 can be combined together for a better comparison purpose. Authors should comment on the repeatability of the observed crack patterns from the DEM simulations with possible variation in the particle arrangement.
11. Figure 18 caption should be checked.
12. The tables should have proper numbering and caption.
13. Some of the references are incomplete and/or erroneous: "Cundall PA, Strack ODJg (1979)", "SCHMIDT GA (2004)", Zhiqi Guo¹, Mark Chapman³ and Xiangyang Li (2012).
14. Though the manuscript is in general well written, some of the sentences are incomplete (line 60) and the manuscript should be checked thoroughly for such errors.

Review form: Reviewer 2 (Arghya Das)

Is the manuscript scientifically sound in its present form?

Yes

Are the interpretations and conclusions justified by the results?

Yes

Is the language acceptable?

Yes

Do you have any ethical concerns with this paper?

No

Have you any concerns about statistical analyses in this paper?

No

Recommendation?

Major revision is needed (please make suggestions in comments)

Comments to the Author(s)

The authors present a numerical study on the effect of cementation strength, gravel content and gravel shape on the deformation and failure characteristics of conglomerate. The authors' main focus is to analyse the crack propagation in the conglomerate under a different set of conditions for the purpose of design of hydraulic fracturing in conglomerate. However, the manuscript needs a lot more effort and it is not properly framed having a lot of mistakes. The following observations need to be addressed carefully.

1. In the second paragraph, the references provided about the studies done on conglomerate are not enough. Please provide some more references to highlight the results and shortcomings of previous researches to make your study more appealing. In brief, the whole introduction part needs to be more elaborated.
2. Please check the manuscript thoroughly for typing error (e.g., Line 57 "duo" should be replaced with "due").
3. Line 88: It is wrong to say that the stress-strain relationship of the parallel bond is similar to that between the particles. The stress-strain behaviour of contacting particles depends on the contact model chosen. Also, note that the authors have not mentioned the contact model chosen for the simulations.
4. Fig 6: The results of the experiment and numerical simulation are not in good agreement as opposed to what the authors report. The authors didn't calibrate the conglomerate numerical model with the experiment model instead calibrated the properties of an individual material (matrix and gravels). Please explain why. Also, mention how the individual properties of matrix and gravels are calibrated.
5. The manuscript is full of grammatical errors. Please check it thoroughly (e.g., Line 46 "Recent years", Line 165 "while keep the other parameters unchanged", Line 260 "As figure 18 shows, the four parameters all change regularly", etc.)
6. Please avoid the use of statement like "was/were", "will" in the manuscript (e.g., Line 73 "We will establish the conglomerate numerical models", Line 188 "mechanical parameters were shown in Table 1, the models were shown in Figure 19", Line 351 "A discrete element numerical model of the conglomerate was established", etc.)
7. The statement "the red lines represent tensile cracks while the green lines represent shear cracks" has been used more than once in the manuscript. Please correct it.

8. Line 232: In the manuscript, Figure 11 has been referred, however, in actual it should be referred as Figure 13.
9. Fig 18: The figure caption is not correct (the lines are directly copied from the manuscript). Please write it correctly.
10. Section 5.2: The procedure for finding gravel critical content (G_{cc}) is not clearly understood. Will G_{cc} depend on the type of packing arrangement? What is the purpose of taking two samples (i.e $R = 5\text{mm}$ & $R = 5.5\text{mm}$) for calculating G_{cc} ? Why a row of particles is removed and not the random particles removed from the sample to obtain G_{cc} ?
11. Please check the formatting of references.

Review form: Reviewer 3 (Aditya Singh)

Is the manuscript scientifically sound in its present form?

Yes

Are the interpretations and conclusions justified by the results?

Yes

Is the language acceptable?

Yes

Do you have any ethical concerns with this paper?

No

Have you any concerns about statistical analyses in this paper?

No

Recommendation?

Major revision is needed (please make suggestions in comments)

Comments to the Author(s)

In the view of this reviewer, the numerical modeling details are not clear in the article. Fundamental information such as boundary conditions, model equations are missing in the article. It is been advised to the authors to add them to the revised article. Otherwise, the article data cannot be reproduced.

Decision letter (RSOS-202178.R0)

Dear Dr Wang

The Editors assigned to your paper RSOS-202178 "Numerical Simulation Study on the Crack Propagation of Conglomerate" have now received comments from reviewers and would like you to revise the paper in accordance with the reviewer comments and any comments from the Editors. Please note this decision does not guarantee eventual acceptance.

Please submit your revised manuscript and required files (see below) no later than 21 days from today's (ie 26-Mar-2021) date. Note: the ScholarOne system will 'lock' if submission of the revision is attempted 21 or more days after the deadline. If you do not think you will be able to meet this deadline please contact the editorial office immediately.

on behalf of Dr Jagdish Shahu (Associate Editor) and R. Kerry Rowe (Subject Editor)
openscience@royalsociety.org

Associate Editor Comments to Author (Dr Jagdish Shahu):

The paper is interesting but several important details are missing. For example, contact model needs to be described, parameter calibration is not clear, and validation with experimental result is not proper. There is also a need to highlight results and shortcomings of previous researches, and to improve overall english.

Reviewer comments to Author:

Reviewer: 1
Comments to the Author(s)

The technical article investigates influence of cementation strength between gravels and matrix, gravel content and shape on the mechanical properties and crack characteristics of conglomerate by applying discrete element based numerical simulation. The manuscript is as such well written and includes interesting observations. However, the reviewer strongly advises to address the following points in a revised submission:

1. Instead of mentioning just the DEM software package, the main governing equations involved in DEM should be presented in Section 2.
2. The contact model employed, i.e. the Parallel Bond Model should be discussed in details with related mathematical equations. Further, the interaction between gravel-gravel particle, matrix-matrix particle and gravel-matrix particle should be addressed with reference to the applied bond model. In Section 3.1 (line no. 117), it should be clearly mentioned how many numbers of clusters were considered to model a typical gravel particle.
3. The procedure of micro-mechanical parameter calibration (Section 3.2) is not very clear. In one place, it has been stated that “In this study, we didn’t correspond the conglomerate model with the laboratory experiment and the micro parameters of matrix and gravels were calibrated respectively” and in other place, it is mentioned that “The micro parameters were finally set to those corresponded to the cementation strength ratio of 0.2 (see in Table 2) by the method of trial and error in which the micro parameters were changed repeatedly. It should be clearly mentioned that the micro-parameters of agglomerates are calibrated against which macro-level data? Further, deviation observed between the experimental result and numerical simulation referred in Fig. 6 should be clearly explained.
4. It is not clear how the crack and cumulative crack is getting defined in the conducted simulations.
5. The cumulative cracks referred in Fig. 9(b) are at which strain level? Similarly, the strain level for the crack patterns shown in Fig. 10, 14, 19 and 20 should be also mentioned.
6. What is crack point strength?
7. At what strain level the elastic modulus referred in Fig. 9(d) has been measured? The initial elastic tangent modulus values for different strength ratio as shown in Fig. 8 seem very similar.
8. What is the definition of the term fractal dimension and how is it getting calculated in Fig. 11?
9. It is not clear how various special shapes gravels are getting characterized in the DEM simulations in comparison to the spherical shaped one. The lack of such shape characterisation might have resulted into high scatter in the results reported in Fig. 13(a-c). The same might be the reason behind not observing any distinct trend in Figure 16.
10. Figure 14 and 15 can be combined together for a better comparison purpose. Authors should comment on the repeatability of the observed crack patterns from the DEM simulations with possible variation in the particle arrangement.
11. Figure 18 caption should be checked.
12. The tables should have proper numbering and caption.
13. Some of the references are incomplete and/or erroneous: “Cundall PA, Strack ODJg (1979)”, “SCHMIDT GA (2004)”, Zhiqi Guo¹, Mark Chapman³ and Xiangyang Li (2012).
14. Though the manuscript is in general well written, some of the sentences are incomplete (line 60) and the manuscript should be checked thoroughly for such errors.

Reviewer: 2

Comments to the Author(s)

The authors present a numerical study on the effect of cementation strength, gravel content and gravel shape on the deformation and failure characteristics of conglomerate. The authors’ main focus is to analyse the crack propagation in the conglomerate under a different set of conditions for the purpose of design of hydraulic fracturing in conglomerate. However, the manuscript needs a lot more effort and it is not properly framed having a lot of mistakes. The following observations need to be addressed carefully.

1. In the second paragraph, the references provided about the studies done on conglomerate are not enough. Please provide some more references to highlight the results and shortcomings of previous researches to make your study more appealing. In brief, the whole introduction part needs to be more elaborated.

2. Please check the manuscript thoroughly for typing error (e.g., Line 57 “duo” should be replaced with “due”).
3. Line 88: It is wrong to say that the stress-strain relationship of the parallel bond is similar to that between the particles. The stress-strain behaviour of contacting particles depends on the contact model chosen. Also, note that the authors have not mentioned the contact model chosen for the simulations.
4. Fig 6: The results of the experiment and numerical simulation are not in good agreement as opposed to what the authors report. The authors didn't calibrate the conglomerate numerical model with the experiment model instead calibrated the properties of an individual material (matrix and gravels). Please explain why. Also, mention how the individual properties of matrix and gravels are calibrated.
5. The manuscript is full of grammatical errors. Please check it thoroughly (e.g., Line 46 “Recent years”, Line 165 “while keep the other parameters unchanged”, Line 260 “As figure 18 shows, the four parameters all change regularly”, etc.)
6. Please avoid the use of statement like “was/were”, “will” in the manuscript (e.g., Line 73 “We will establish the conglomerate numerical models”, Line 188 “mechanical parameters were shown in Table 1, the models were shown in Figure 19”, Line 351 “A discrete element numerical model of the conglomerate was established”, etc.)
7. The statement “the red lines represent tensile cracks while the green lines represent shear cracks” has been used more than once in the manuscript. Please correct it.
8. Line 232: In the manuscript, Figure 11 has been referred, however, in actual it should be referred as Figure 13.
9. Fig 18: The figure caption is not correct (the lines are directly copied from the manuscript). Please write it correctly.
10. Section 5.2: The procedure for finding gravel critical content (Gcc) is not clearly understood. Will Gcc depend on the type of packing arrangement? What is the purpose of taking two samples (i.e R = 5mm & R = 5.5mm) for calculating Gcc ? Why a row of particles is removed and not the random particles removed from the sample to obtain Gcc?
11. Please check the formatting of references.

Reviewer: 3

Comments to the Author(s)

In the view of this reviewer, the nuclear modeling details are not clear in the article. boundary conditions, model details are missing in the article. It is been advised to the authors to add them to the revised article. Otherwise, the article data cannot be repeated.

===PREPARING YOUR MANUSCRIPT===

Your revised paper should include the changes requested by the referees and Editors of your manuscript. You should provide two versions of this manuscript and both versions must be provided in an editable format:
 one version identifying all the changes that have been made (for instance, in coloured highlight, in bold text, or tracked changes);
 a 'clean' version of the new manuscript that incorporates the changes made, but does not highlight them. This version will be used for typesetting if your manuscript is accepted.
 Please ensure that any equations included in the paper are editable text and not embedded images.

Please ensure that you include an acknowledgements' section before your reference list/bibliography. This should acknowledge anyone who assisted with your work, but does not

qualify as an author per the guidelines at <https://royalsociety.org/journals/ethics-policies/openness/>.

===PREPARING YOUR REVISION IN SCHOLARONE===

-- Ensure that your data access statement meets the requirements at <https://royalsociety.org/journals/authors/author-guidelines/#data>. You should ensure that you cite the dataset in your reference list. If you have deposited data etc in the Dryad repository, please include both the 'For publication' link and 'For review' link at this stage.

Author's Response to Decision Letter for (RSOS-202178.R0)

See Appendix A.

RSOS-202178.R1 (Revision)

Review form: Reviewer 1

Is the manuscript scientifically sound in its present form?

Yes

Are the interpretations and conclusions justified by the results?

Yes

Is the language acceptable?

Yes

Do you have any ethical concerns with this paper?

No

Have you any concerns about statistical analyses in this paper?

No

Recommendation?

Accept with minor revision (please list in comments)

Comments to the Author(s)

Most of the comments given by the reviewer has been addressed in the revised submission. However, following points should also be addressed:

1. In Section-2, what do the symbols R and t stands for in Eqs. 1, 2 and 3?
2. Equation 5 should be a vector summation.
3. In Eqs, 6 and 7, why the parallel bond cross-sectional area term is appearing and how is it getting calculated during the calibration process?
4. Table-1 lists out only the normal to shear stiffness ratios for the particle contacts and the parallel bond model, it still not clear what are the adopted value of the stiffness and how is it getting calculated (no equation related to contact stiffness model has been referred in Section-2).
5. Is the unit direction vector referred in Eq. 8 is along the contact normal? There exists some typo in the manuscript while writing the unit direction vector symbol.
6. What are the references in regard to the adopted parameter values of uniaxial compressive strength (UCS) and elastic modulus of andesite in Section 3.2? In addition what are the references related to the micro indentation process mentioned in Section 3.2 and the rational behind adopted average hardness ratio (HR) and elastic modulus ratio (ER) values for the matrix to the gravel properties?
7. The reference style of Cundall and Strack (1979) should be checked.

Review form: Reviewer 2 (Arghya Das)

Is the manuscript scientifically sound in its present form?

Yes

Are the interpretations and conclusions justified by the results?

Yes

Is the language acceptable?

Yes

Do you have any ethical concerns with this paper?

No

Have you any concerns about statistical analyses in this paper?

No

Recommendation?

Accept as is

Comments to the Author(s)

The authors addressed the comments of the reviewers.

Review form: Reviewer 3 (Aditya Singh)

Is the manuscript scientifically sound in its present form?

Yes

Are the interpretations and conclusions justified by the results?

Yes

Is the language acceptable?

Yes

Do you have any ethical concerns with this paper?

No

Have you any concerns about statistical analyses in this paper?

No

Recommendation?

Accept as is

Comments to the Author(s)

The manuscript is acceptable.

Decision letter (RSOS-202178.R1)

Dear Dr Wang,

It is a pleasure to accept your manuscript entitled "Numerical Simulation Study on the Crack Propagation of Conglomerate" in its current form for publication in Royal Society Open Science.

The comments of the reviewer(s) who reviewed your manuscript are included at the foot of this letter.

on behalf of Dr Jagdish Shahu (Associate Editor) and R. Kerry Rowe (Subject Editor)
openscience@royalsociety.org

Associate Editor Comments to Author (Dr Jagdish Shahu):

All three reviewers have now recommended acceptance of the manuscript. However, Reviewer 1 has noted few minor discrepancies. The authors should make the suggested revisions within their manuscript proofs once they receive them.

Reviewer comments to Author:

Reviewer: 1

Comments to the Author(s)

Most of the comments given by the reviewer has been addressed in the revised submission.

However, following points should also be addressed:

1. In Section-2, what do the symbols R and t stands for in Eqs. 1, 2 and 3?
2. Equation 5 should be a vector summation.
3. In Eqs, 6 and 7, why the parallel bond cross-sectional area term is appearing and how is it getting calculated during the calibration process?
4. Table-1 lists out only the normal to shear stiffness ratios for the particle contacts and the parallel bond model, it still not clear what are the adopted value of the stiffness and how is it getting calculated (no equation related to contact stiffness model has been referred in Section-2).
5. Is the unit direction vector referred in Eq. 8 is along the contact normal? There exists some typo in the manuscript while writing the unit direction vector symbol.
6. What are the references in regard to the adopted parameter values of uniaxial compressive strength (UCS) and elastic modulus of andesite in Section 3.2? In addition what are the references related to the micro indentation process mentioned in Section 3.2 and the rational behind adopted average hardness ratio (HR) and elastic modulus ratio (ER) values for the matrix to the gravel properties?
7. The reference style of Cundall and Strack (1979) should be checked.

Reviewer: 2

Comments to the Author(s)

The authors addressed the comments of the reviewers.

Reviewer: 3

Comments to the Author(s)

The manuscript is acceptable.

Appendix A

Reviewer: 1

Comments to the Author(s)

The technical article investigates influence of cementation strength between gravels and matrix, gravel content and shape on the mechanical properties and crack characteristics of conglomerate by applying discrete element based numerical simulation. The manuscript is as such well written and includes interesting observations. However, the reviewer strongly advises to address the following points in a revised submission:

1. Instead of mentioning just the DEM software package, the main governing equations involved in DEM should be presented in Section 2.

--The parallel bond model is used in this paper, and the main governing equations have been supplemented in Section 2.

'Specifically, the movement between particles is determined by Newton's second law, while the interaction between particles is determined by the force-displacement law. The force-displacement law of force and moment in the parallel bond model can be illustrated as follow.

1. Update properties on the bond cross-section:

$$\bar{R} = \bar{\lambda} \begin{cases} \min(R^{(1)}, R^{(2)}), & \text{ball-ball} \\ R^{(1)}, & \text{ball-facet} \end{cases} \quad (1) \quad \bar{A} = \begin{cases} 2\bar{R}t, & 2D(t=1) \\ \pi\bar{R}^2, & 3D \end{cases} \quad (2)$$

$$\bar{I} = \begin{cases} \frac{2}{3}t\bar{R}^3, & 2D(t=1) \\ \frac{1}{4}\pi\bar{R}^4, & 3D \end{cases} \quad (3) \quad \bar{J} = \begin{cases} 0, & 2D \\ \frac{1}{2}\pi\bar{R}^4, & 3D \end{cases} \quad (4)$$

Where $\bar{\lambda}$ is radius multiplier, which is 1.0 by default. \bar{A} is the parallel bond cross-sectional area. \bar{I} is the inertia moment on the bond cross-section. \bar{J} is the polar inertia moment on the bond cross-section.

2. Update the parallel-bond force:

$$\vec{\bar{F}} = \vec{\bar{F}}_n + \vec{\bar{F}}_s \quad (5)$$

$$\bar{F}_n := \bar{F}_n + \bar{k}_n \bar{A} \Delta\delta_n \quad (6)$$

$$\bar{F}_s := \bar{F}_s + \bar{k}_s \bar{A} \Delta\delta_s \quad (7)$$

As shown in formula (5), the parallel-bond force between particles is decomposed into a normal force and a shear force. Equation (6) and (7) are the updated parallel-bond force in each direction. \bar{k}_n and \bar{k}_s is the normal and shear stiffness, respectively. $\Delta\delta_n$ and $\Delta\delta_s$ is the relative increment of normal-displacement and shear-displacement, respectively.

3. Update the parallel-bond moment:

$$\bar{M} = \bar{M}_t n_c + \bar{M}_b \quad (2D \text{ model} : \bar{M}_t \equiv 0) \quad (8)$$

$$\bar{M}_t := \begin{cases} 0, & 2D \\ \bar{M}_t - \bar{k}_s \bar{J} \Delta \theta_t, & 3D \end{cases} \quad (9)$$

$$\bar{M}_b := \bar{M}_b - \bar{k}_n \bar{I} \Delta \theta_b \quad (10)$$

As shown in formula (8), the parallel-bond moment is decomposed into a torque and a bending moment. n_c is the unit direction vector. Equation (9) and (10) are the updated torque and bending moment. θ_t and θ_b is the relative increment of torsion angle and bend-rotation, respectively.

4. Update the normal and shear stresses at the parallel-bond periphery:

$$\bar{\sigma} = \frac{\bar{F}_n}{\bar{A}} + \bar{\beta} \frac{\|\bar{M}_b\|}{\bar{I}} \quad (11)$$

$$\bar{\tau} = \frac{\|\bar{F}_s\|}{\bar{A}} + \begin{cases} 0, & 2D \\ \bar{\beta} \frac{|\bar{M}_t| \bar{R}}{\bar{J}}, & 3D \end{cases} \quad (12)$$

$$\text{With } \bar{\beta} \in [0,1]$$

Where $\bar{\sigma}$ and $\bar{\tau}$ are the normal stress and shear stress at the parallel-bond periphery, respectively. $\bar{\beta}$ is the moment-contribution factor, ranging from 0~1.

5. Bond failure criteria between particles:

While the maximum normal stress between particles is greater than the tensile strength ($\bar{\sigma} > \bar{\sigma}_c$), tensile failure occurs. Shear strength $\bar{\tau}_c = c + \sigma \tan \bar{\phi}$. $\sigma = \bar{F}_n / \bar{A}$ is the average normal stress on the parallel bond cross-section. While the maximum shear stress between particles is greater than the shear strength ($\bar{\tau} > \bar{\tau}_c$), shear failure occurs.'

2. The contact model employed, i.e. the Parallel Bond Model should be discussed in details with related mathematical equations. Further, the interaction between gravel-gravel particle, matrix-matrix particle and gravel-matrix particle should be addressed with reference to the applied bond model. In Section 3.1 (line no. 117), it should be clearly mentioned how many numbers of clusters were considered to model a typical gravel particle.

--The mathematical equations related to the parallel bond model have been supplemented in Section 2. In addition, the interaction between gravel-gravel particle, matrix-matrix particle and gravel-matrix particle are all applied parallel bond model. there are 26 typical clusters in the model. For the cluster mentioned in line 117, this paper adds that there are 26 typical clusters in Figure 4

'As shown in Figure 4, there are 26 typical clusters in this model'

3. The procedure of micro-mechanical parameter calibration (Section 3.2) is not very

clear. In one place, it has been stated that “In this study, we didn’t correspond the conglomerate model with the laboratory experiment and the micro parameters of matrix and gravels were calibrated respectively” and in other place, it is mentioned that “The micro parameters were finally set to those corresponded to the cementation strength ratio of 0.2 (see in Table 2) by the method of trial and error in which the micro parameters were changed repeatedly. It should be clearly mentioned that the micro-parameters of agglomerates are calibrated against which macro-level data? Further, deviation observed between the experimental result and numerical simulation referred in Fig. 6 should be clearly explained.

--We have modified the micromechanical parameter calibration (subsection 3.2) to make the calibration procedure clearer. The experimental results are compared with the numerical results

‘Since there is no specific correspondence between the micro parameters of particles in PFC and the macro mechanical properties of the rocks, we adopted the method of trial and error to change the micro parameters repeatedly till they met with the macro mechanical properties of the material. In the conglomerate sample as shown in Figure 2, the gravel is mainly andesite, and matrix is formed from the compaction and cementation of minerals such as quartz, feldspar, calcite and clay, while the cementation between gravel and matrix is mainly argillaceous cementing, average strength from high to low of which is: gravel>matrix>cementing between gravel and matrix. To simplify the numerical model, we assume that the gravel and the matrix are single substance, and the macro mechanical parameters of andesite are used to calibrate the micro mechanical parameters of the gravel. The uniaxial compressive strength (UCS) of andesite is 175MPa, while the elastic modulus (E) is 69.98GPa.

Based on the conglomerate sample as shown in Fig. 2(a), we established the numerical sample (see Fig. 4), and obtained the average hardness and average elastic modulus of the gravel and matrix according to the micro indentation. The calculated average hardness ratio (HR) of matrix to gravel is 0.08, and the average elastic modulus ratio (ER) of matrix to gravel is 0.27. Finally, we calibrated the conglomerate micro mechanical parameters through the uniaxial compression test. Specific calibration flow chart is shown in Figure 5.

Through repeated trial and error, the micro mechanical parameters of gravel, matrix and cementing surface are determined, as shown in Table 1 and Table 3, respectively (the cementation strength ratio is set to 0.2). Figure 6 shows the uniaxial compressive stress-strain and cumulative crack number curves of gravel and matrix. The final calibration results are as follows, the uniaxial compressive strength (UCS) of gravel is 175MPa, the elastic modulus is 69.98GPa, the uniaxial compressive strength (UCS) of the matrix sample is 110MPa, while the elastic modulus is 13.19GPa. Comparison between the mechanical parameters gained through the conglomerate numerical model and laboratory test is shown in Table 2. The errors of compressive strength and elastic modulus are 9.42% and 4.93%, respectively. The final failure characteristics (see Fig. 7) are in good agreement with the test results (Fig. 7, b). The

red lines represent tensile cracks while the green lines represent shear cracks. Therefore, the selection of micro mechanical parameters is reasonable.'

Figure 5 Flow chart of parameter calibration for conglomerate numerical model

Table 2 Comparison of mechanical parameters between numerical model and laboratory test

Parameter	Numerical model	Test sample	Error/%
UCS/MPa	66.2	60.5	9.42
E/GPa	19.17	18.27	4.93

Figure 7 Failure mode of complete conglomerate sample gained from (a) numerical simulation and (b) experiment.

4. It is not clear how the crack and cumulative crack is getting defined in the conducted simulations.

--We have supplemented the definitions of crack and cumulative crack in this paper. 'Once the maximum principal stress is higher than the corresponding bond strength, the parallel bond will be broken. Crack will be generated between the particles, the bond, force and torque at this position will then be removed. Under an external loading, the bond continues to break, and the total number of cracks gradually

increases, which is the cumulative crack.'

5. The cumulative cracks referred in Fig. 9(b) are at which strain level? Similarly, the strain level for the crack patterns shown in Fig. 10, 14, 19 and 20 should be also mentioned.

--The cumulative cracks in Fig. 9(b), 13(b) and 18(b) are calculated when the numerical model stops loading, and correspond to Fig 10, 15 and 20 respectively. In this paper, Fig 14 and Fig 15 show the macroscopic failure of the test in different ways when the loading stops: Figure 14 shows the position of the particles, but Figure 15 only shows the mode of crack. Fig. 19 and Fig. 20 are also described above. We also added the corresponding strain levels for Fig 10, 15 and 20

Figure 10 Crack shapes under different cementation strength ratios (In the figure, ϵ_a is the axial strain corresponding to the fracture shape when the load is stopped).

Figure 15 Crack shape under different special-shaped gravel contents.

Figure 20 Crack shapes under different spherical gravel contents.

6. What is crack point strength?

--We have added the definition

‘Crack initiation point strength is the axial stress corresponding to the first crack occurred in the conglomerate numerical model in uniaxial compression test. Therefore, crack initiation point strength/UCS is an important parameter to evaluate the initial crack time inside the rock. The total number of cracks is the number of all the cracks in the model after the uniaxial compression test, that is, the number of all the cracks in the corresponding crack shape map.’

7. At what strain level the elastic modulus referred in Fig. 9(d) has been measured? The initial elastic tangent modulus values for different strength ratio as shown in Fig. 8 seem very similar.

--All the elastic moduli in this paper are tangent moduli. The specific calculation method is as follows: take 50% of the peak value of the stress-strain curve to calculate the tangent slope. This is why the initial moduli in Fig. 8 are very similar, but there are differences in statistics in Fig. 9(d).

8. What is the definition of the term fractal dimension and how is it getting calculated in Fig. 11?

-- We add the definition of fractal dimension and the calculation process

‘The fractal theory could reflect the effectiveness of the space that occupied by complex shapes, it is a measure of the irregularities of complex shapes. Therefore in this paper, we used fractal dimension to quantitatively evaluate the complexity of crack morphology. The fractal dimension D is given as follows:

$$D = \lim_{\varepsilon \rightarrow 0} \left[\log N(\varepsilon) / \log(1/\varepsilon) \right] \quad (13)$$

In this paper, a small cube with side length ε is used to cover the cracks, and $N(\varepsilon)$ is the number of small cubes required to cover the cracks. The specific calculation is done using the FracLab toolbox in Matlab. First, FracLab automatically generates

the upper and lower limits of ε according to the size of the picture. Then, choose how many small cubes of different sizes to generate. Different sizes of ε correspond to different $N(\varepsilon)$. The progression of ε can be power law or linear. Finally, draw a scatter plot with x-axis $\log_2(\varepsilon)$ and y-axis $\log_2(N)$ in a two-dimensional coordinate system. Fit the scatter plot into a straight line, and the opposite number of the slope of the straight line is the fractal dimension D .'

9. It is not clear how various special shapes gravels are getting characterized in the DEM simulations in comparison to the spherical shaped one. The lack of such shape characterisation might have resulted into high scatter in the results reported in Fig. 13(a-c). The same might be the reason behind not observing any distinct trend in Figure 16.
-- At present, we have not found a method to characterize the gravel with various special shapes. Only the problems of different cementation and different content are studied, and further research will be carried out for different shapes of gravel in the future.

10. Figure 14 and 15 can be combined together for a better comparison purpose. Authors should comment on the repeatability of the observed crack patterns from the DEM simulations with possible variation in the particle arrangement.
--The influence of different spatial arrangement of gravels on crack propagation is also the next problem to be studied.

11. Figure 18 caption should be checked.
-- We have modified.

12. The tables should have proper numbering and caption.
-- We have modified.

13. Some of the references are incomplete and/or erroneous: "Cundall PA, Strack ODJg (1979)", "SCHMIDT GA (2004)", Zhiqi Guo¹, Mark Chapman³ and Xiangyang Li (2012).
-- We have modified.

14. Though the manuscript is in general well written, some of the sentences are incomplete (line 60) and the manuscript should be checked thoroughly for such errors.
-- We have modified.

Reviewer: 2

Comments to the Author(s)

The authors present a numerical study on the effect of cementation strength, gravel content and gravel shape on the deformation and failure characteristics of conglomerate. The authors' main focus is to analyse the crack propagation in the conglomerate under a different set of conditions for the purpose of design of hydraulic fracturing in conglomerate. However, the manuscript needs a lot more effort and it is not properly framed having a lot of mistakes. The following observations need to be addressed carefully.

1. In the second paragraph, the references provided about the studies done on conglomerate are not enough. Please provide some more references to highlight the results and shortcomings of previous researches to make your study more appealing. In brief, the whole introduction part needs to be more elaborated.

-- *We have modified.*

' In addition, there were few studies on influences of particle size on mechanical properties of rocks. Olsson studied and calculated the yield stress of marble with a particle size of 0.005~3.5mm, and believed that the yield stress was negatively correlated with particle size in a nonlinear manner (15). Conglomerate with particle size up to 30mm also showed a similar rule. The larger the gravel particle size, the stronger the heterogeneity of particle size, the lower the compressive strength and Young's modulus, and the more obvious the plastic characteristics of stress-strain curve (16-19). Conglomerate is generally regarded as a two-phase composite material due to the great difference in mechanical properties between gravel and matrix. A research on gypsum and concrete also conformed to the characteristics of two-phase composite materials. An uniaxial compression results of artificial composite samples showed that the compressive strength is negatively correlated with the content of hard particles, and the stress concentration at the edge of hard particles led to cracking around hard particles (20, 21) The ice-rock mixture is also a two-phase combination, and a great deal of research has been carried out to study the mechanical properties of it in ice sheets or glaciers. When the content of rock particles was small (about 6~15% according to different studies), the ice-rock mixture behaved as pure ice, and became hardened and stronger as the proportion of mineral particles increased. When the content increased further ($\geq 56\%$), the strength of the mixed phase was close to that of dry sand(22-24). The law was similar to the strengthening effect of adding granular materials in metal materials(25-28). Chao Qi et al. believed that the hard particles blocked the lattice dislocation and increased the overall strength of the composite phase(29). This was not consistent with the laws of rock, gypsum and concrete. All studies above have not systematically studied influences of gravel particle size and volume fraction on the brittle-plasticity and mechanical properties of rocks, which will be discussed in this paper. '

2. Please check the manuscript thoroughly for typing error (e.g., Line 57 "duo" should be replaced with "due").

-- *We have modified.*

3. Line 88: It is wrong to say that the stress-strain relationship of the parallel bond is similar to that between the particles. The stress-strain behaviour of contacting particles depends on the contact model chosen. Also, note that the authors have not mentioned the contact model chosen for the simulations.

--*We have deleted this paragraph that the stress-strain relationship of the parallel bond is similar to that between the particles. In addition, the use of parallel bond model to build conglomerate model is mentioned in Section 3.1.*

‘We conducted this study on conglomerate and applied PBM (Parallel Bond Model) contact model to establish its discrete element model and study its mechanical characteristics and laws, based on the PFC2D particle flow software platform combining its geometric and mechanical characteristics.’

4. Fig 6: The results of the experiment and numerical simulation are not in good agreement as opposed to what the authors report. The authors didn’t calibrate the conglomerate numerical model with the experiment model instead calibrated the properties of an individual material (matrix and gravels). Please explain why. Also, mention how the individual properties of matrix and gravels are calibrated.

--*We have modified the micromechanical parameter calibration (subsection 3.2) to make the calibration procedure clearer.*

‘Since there is no specific correspondence between the micro parameters of particles in PFC and the macro mechanical properties of the rocks, we adopted the method of trial and error to change the micro parameters repeatedly till they met with the macro mechanical properties of the material. In the conglomerate sample as shown in Figure 2, the gravel is mainly andesite, and matrix is formed from the compaction and cementation of minerals such as quartz, feldspar, calcite and clay, while the cementation between gravel and matrix is mainly argillaceous cementing, average strength from high to low of which is: gravel>matrix>cementing between gravel and matrix. To simplify the numerical model, we assume that the gravel and the matrix are single substance, and the macro mechanical parameters of andesite are used to calibrate the micro mechanical parameters of the gravel. The uniaxial compressive strength (UCS) of andesite is 175MPa, while the elastic modulus (E) is 69.98GPa.

Based on the conglomerate sample as shown in Fig. 2(a), we established the numerical sample (see Fig. 4), and obtained the average hardness and average elastic modulus of the gravel and matrix according to the micro indentation. The calculated average hardness ratio (HR) of matrix to gravel is 0.08, and the average elastic modulus ratio (ER) of matrix to gravel is 0.27. Finally, we calibrated the conglomerate micro mechanical parameters through the uniaxial compression test. Specific calibration flow chart is shown in Figure 5.

Through repeated trial and error, the micro mechanical parameters of gravel, matrix and cementing surface are determined, as shown in Table 1 and Table 3, respectively

(the cementation strength ratio is set to 0.2). Figure 6 shows the uniaxial compressive stress-strain and cumulative crack number curves of gravel and matrix. The final calibration results are as follows, the uniaxial compressive strength (UCS) of gravel is 175MPa, the elastic modulus is 69.98GPa, the uniaxial compressive strength (UCS) of the matrix sample is 110MPa, while the elastic modulus is 13.19GPa. Comparison between the mechanical parameters gained through the conglomerate numerical model and laboratory test is shown in Table 2. The errors of compressive strength and elastic modulus are 9.42% and 4.93%, respectively. The final failure characteristics (see Fig. 7) are in good agreement with the test results (Fig. 7, b). The red lines represent tensile cracks while the green lines represent shear cracks. Therefore, the selection of micro mechanical parameters is reasonable.’

Figure 5 Flow chart of parameter calibration for conglomerate numerical model

Table 2 Comparison of mechanical parameters between numerical model and laboratory test

Parameter	Numerical model	Test sample	Error/%
UCS/MPa	66.2	60.5	9.42
E/GPa	19.17	18.27	4.93

Figure 7 Failure mode of complete conglomerate sample gained from (a) numerical simulation and (b) experiment.

(a)

(b)

5. The manuscript is full of grammatical errors. Please check it thoroughly (e.g., Line 46 “Recent years”, Line 165 “while keep the other parameters unchanged”, Line 260 “As figure 18 shows, the four parameters all change regularly”, etc.)

-- *We have modified.*

6. Please avoid the use of statement like “was/were”, “will” in the manuscript (e.g., Line 73 “We will establish the conglomerate numerical models”, Line 188 “mechanical parameters were shown in Table 1, the models were shown in Figure 19”, Line 351 “A discrete element numerical model of the conglomerate was established”, etc.)

-- *We have modified.*

7. The statement “the red lines represent tensile cracks while the green lines represent shear cracks” has been used more than once in the manuscript. Please correct it.

-- *We have modified.*

8. Line 232: In the manuscript, Figure 11 has been referred, however, in actual it should be referred as Figure 13.

-- *We have modified.*

9. Fig 18: The figure caption is not correct (the lines are directly copied from the manuscript). Please write it correctly.

-- *We have modified.*

10. Section 5.2: The procedure for finding gravel critical content (G_{cc}) is not clearly understood. Will G_{cc} depend on the type of packing arrangement? What is the purpose of taking two samples (i.e $R = 5\text{mm}$ & $R = 5.5\text{mm}$) for calculating G_{cc} ? Why a row of particles is removed and not the random particles removed from the sample to obtain G_{cc} ?

--*A strong force chain doesn't form in the entire sample when the conglomerate is*

supported by matrix. Thus, to get the G_{cc} corresponding to the changing point of support pattern, the dotted spherical gravels are removed from the model (Figure 22(b), 23(b)). If the particles are removed randomly, for example, only one particle is removed, the strong chain will still form between the gravels. However, the removal of a row of particles ensures that there is no complete chain of strength between the gravels.

‘A strong force chain is formed throughout the gravels under uniaxial compression. Yet, it wasn’t formed in the entire sample when the conglomerate is supported by matrix. Thus, to get the G_{cc} corresponding to the changing point of support pattern, the dotted spherical gravels were removed from the model (Figure 22(b), 23(b)).’

11. Please check the formatting of references.

--We have modified it, and the format is Vancouver style.

Reviewer: 3

Comments to the Author(s)

In the view of this reviewer, the nuclear modeling details are not clear in the article. boundary conditions, model details are missing in the article. It is been advised to the authors to add them to the revised article. Otherwise, the article data cannot be repeated.

-- The boundary condition is that the model is bound by the upper and lower walls, and the model is loaded by moving the upper and lower walls. The modeling details are mentioned in Section 3.1.

‘During the uniaxial compression test, load was applied through applying displacements on the top and bottom walls and the displacement rate applied by the device should be sufficiently slow to ensure the loading is quasi-static(9). The loading rate in this study was 0.025m/s, and the sample lost stability gradually and cracked as the axial pressure increased. Tests conducted in Section 4.1 and 4.2 were both stopped when the axial stress was 60% of the peak strength, while to speed up the progress, the test conducted in Section 4.3 was stopped when the axial stress was 70% of the peak strength.’

‘To make the model more accurate, taking Figure 3(a) as example, we first unfolded the outer surface of the cylindrical sample (Figure 3) as shown in Figure 3(b), then selected a region of 50×100 mm represented by the black dotted box in Figure 3(b) to pick the gravel shapes, sizes and distribution and obtain the digital images which were finally imported into the PFC2D software to establish the 2D model as shown in Figure 3(c). Based on the conglomerate as shown in Figure 1, we established a series of different models. To build the conglomerate model with different gravel contents, we took Figure 3(c) as the basic model and removed some of the particles to obtain the conglomerate model with gravel volume contents ranging from 20% to 35% (area percent in the model). Meanwhile, we established the conglomerate models with 55% and 75% gravel volume contents based on Figure 2(d) and 2(e) as shown in Figure 3(d).’